

# 1 Characterization of atmospheric trace gases and particle matters in
# 2 Hangzhou, China

Gen Zhang[1], Honghui Xu[2], Bing Qi[3], Rongguang Du[3], Ke Gui[1], Hongli Wang[4], Wanting Jiang[5], Linlin
Liang[1], Wanyun Xu[1]
[1]State Key Laboratory of Severe Weather & Key Laboratory of Atmospheric Chemistry of CMA, Chinese Academy of
Meteorological Sciences, Beijing 100081, China
[2]Zhejiang Institute of Meteorological Science, Hangzhou 310008, China
[3]Hangzhou Meteorological Bureau, Hangzhou 310051, China
[4]State Environmental Protection Key Laboratory of Formation and Prevention of Urban Air Pollution Complex, Shanghai
Academy of Environmental Sciences, Shanghai 200233, China
[5]Plateau Atmospheric and Environment Laboratory of Sichuan Province, College of Atmospheric Science, Chengdu
University of Information Technology, Chengdu 610225, China
*Correspondence to*: Gen Zhang (zhangg@camscma.cn) and Honghui Xu (forsnow@126.com)
**Abstract.** The Yangtze River Delta (YRD) is one of the most densely populated regions in China with severe air quality
issues, which has not been fully understood. Thus, in this study, based on one-year (2013) continuous measurement at a
National Reference Climatological Station (NRCS, 30.22°N, 120.17°E, 41.7 m a. s. l) in the center of Hangzhou in the YRD,
we investigated the seasonal characteristics, interspecies relationships, and the local emissions and the regional potential
source contributions of trace gases (including $O_3$, $NO_x$, $NO_y$, $SO_2$ and CO) and particulate matters ($PM_{2.5}$ and $PM_{10}$). Results
revealed severe two-tier air pollution (photochemical and haze pollution) occurred in this region, with frequent exceedances
in $O_3$ (38 days) and $PM_{2.5}$ (62 days). $O_3$ and $PM_{2.5}$ both exhibited distinct seasonal variations with reversed patterns: $O_3$
reaching a maximum in warm seasons (May and July) but $PM_{2.5}$ in cold seasons (November to January). The overall results
from interspecies correlation indicated a strong local photochemistry favoring the $O_3$ production under a volatile organic
compound (VOC)-limited regime, whereas it moved towards an optimum $O_3$ production zone during warm seasons,
accompanying with a formation of secondary fine particles under high $O_3$. The emission maps of $PM_{2.5}$, CO, $NO_x$, and $SO_2$
demonstrated that local emissions were both significant for these species on seasonal scale. The contributions from the
regional transports among inland cities (Zhejiang, Jiangsu, Anhui, and Jiangxi Province) on seasonal scale were further
confirmed to be crucial to air pollution at NRCS site by using the backward trajectories simulations. Air masses transported
from Yellow Sea, East Sea, and South Sea were also found to be highly relevant to the elevated pollutants, especially for
$NO_x$ and $O_3$. Case studies of photochemical pollution ($O_3$) and haze ($PM_{2.5}$) episodes both suggested the combined
importance of local atmospheric photochemistry and synoptic conditions during the accumulation (related with anticyclones)
and dilution process (related with cyclones). This study supplements a general picture of the air pollution state in the YRD



region, specifically elucidates the role of local emission and regional transport, and interprets the physical and photochemical
processes during haze and photochemical pollution episodes. Moreover, this work suggests that cross-regional control
measures are crucial to improve air quality in the YRD region, and further emphasizes the importance of local thermally
induced circulation on air quality.

## 1 Introduction

Ambient air quality is mainly affected by particle matters and gaseous pollutants such as ozone ($O_3$), nitrogen oxides ($NO_x$),
carbon monoxide (CO), and sulfur dioxide ($SO_2$). Particle matters ($PM_{2.5}$ and $PM_{10}$) are both from natural sources (e.g.,
windborne dust, volcanoes) or anthropogenic activities such as fossil and biomass fuel combustion (Chow and Watson,
2002). They have received extensive attention due to their harmful impact not only on human health such as aggravating
chronic respiratory and cardiovascular diseases (Pope et al., 1999) but also on climate change (Seinfeld et al., 2004; IPCC,
2007; Mercado et al., 2009). As primary gaseous pollutants, $NO_x$, CO, and $SO_2$ are all trace gases and derived from the
anthropogenic activities (Kato and Akimoto, 1994; Streets and Waldhoff, 2000). $NO_x$, with a short lifetime, is mainly
emitted from the fuel burning in the polluted region. In contrast, CO has a relatively long atmospheric lifetime and emitted
from the combustion sources, thus it's also a preferred tracer for indicating the anthropogenic pollution and charactering the
other pollutants (Jaffe et al., 1997; Parrish et al., 1998). In addition to the net downward transport of $O_3$ by eddy diffusion
from the stratosphere aloft, tropospheric $O_3$ is a well-known secondary gaseous pollutants and formed through the
photochemical oxidation of volatile organic compounds (VOCs) and nitrogen oxides ($NO_x$) under the irradiation of sunlight
(Logan, 1985; Roelofs et al., 1997), which has been also increasingly concerned with its adverse effect on exacerbating
chronic respiratory diseases and causing short-term reductions in lung function (Shao et al., 2006; Streets et al., 2007; Liu et
al., 2013) and vegetation (Feng et al., 2014). Reactive nitrogen ($NO_y$) is defined as the sum of $NO_x$ and all compounds that
are products of the atmospheric oxidation of $NO_x$ (e.g., PANs, $HNO_3$, and HONO). Except $NO_x$, the other constituents in
$NO_y$ are also mainly produced via the complex conversions within primary gaseous pollutants (i.e., photochemical oxidation
and nighttime chemistry). Moreover, some critical interactions have been verified existing between the gaseous pollutants
and or particle matters (Zhang et al., 2004; Cheng et al., 2016). For instance, in the presence of high $NH_3$ and low air
temperature, ammonium nitrate ($NH_4NO_3$) is formed in regions with $HNO_3$ and $NH_3$, which is an important constituent of
$PM_{2.5}$ under the high $NO_x$ condition (Seinfeld and Pandis, 2006). To some extent, such interactions further improve or
deteriorate the air quality. The oxidation of $SO_2$ can lead to acid deposition but also contributes to the formation of sulphate
aerosols (Meagher et al., 1978; Saxena and Seigneur, 1987), which in turn will influence the solar radiation and
photochemistry (Dickerson et al., 1997) and further weaken the formation of secondary pollutants. Therefore, clear
understanding in their characteristics, sources, transport, and formation mechanisms including interactions is crucial for
gaining the comprehensive information on the complex air pollution.
The Yangtze River Delta (YRD) region is located in the eastern of China, including the mega-city Shanghai and the well-
industrialized areas of southern Jiangsu Province and northern Zhejiang Province, with over ten large cities such as



Hangzhou, Suzhou, Wuxi and Changzhou lying along the mid-YRD (Fig. 1). Being one of the most rapid growths of
transportation, industries, and urbanization regions in China, it has been became hot spot with air pollution problems over
the past three decades, together with the Pearl River Delta (PRD) and Beijing-Tianjin-Hebei (BTH) region. To date,
numerous combined studies of $O_3$ and $PM_{2.5}$ were implemented in representative urban cities in YRD region such as
Shanghai (Geng et al., 2007; Ding et al., 2013; Li et al., 2016a; Miao et al., 2017a) and Nanjing (Wang et al., 2002; Wang et
al., 2003; Kang et al., 2013; Chen et al., 2016). On the contrary, in Hangzhou (29.25°-30.5 °N, 118.34 °-120.75 °E), a capital
city of Zhejiang Province in YRD region, which is lying along the mid-YRD, only a few sole studies of $PM_{2.5}$ or $O_3$ were
sporadically conducted. To our knowledge, the pioneer measurement of $O_3$ in or around Hangzhou started in the 1990s at
Lin'an site, a regional station located in the east Zhejiang Province (50 km away from Hangzhou) (Luo et al., 2000).
Subsequent studies at this site depicted the first picture of the seasonal variations of $O_3$ and its precursors (Wang et al., 2001;
Wang et al., 2004). Xu et al. (2016) concluded the medium long range boundary transport of air masses coming from
biomass burning regions was responsible for the formation of haze aerosols at Lin'an site during the winter. In the urban
Hangzhou, Li et al (2017) recently reported the results of short-term measurements of $O_3$, CO, and non-methane
hydrocarbons at three sites in Hangzhou in the summertime of 2013. In terms of particle matters, Wu et al. (2016a) reported
that the local vehicle emission was a major contribution to $PM_{2.5}$, while results from Yu et al. (2014) suggested cross-border
transports rather than local emissions control high $PM_{2.5}$ concentration and formation. Hence, large knowledge gap and
discrepancy still exist in understanding the complex combined pollution of $O_3$ and $PM_{2.5}$ in Hangzhou.
To supplement the picture of air pollution in the YRD, we conducted continuous measurements of trace gases ($O_3$, $NO_x$,
$NO_y$, CO, and $SO_2$) and particle matters ($PM_{2.5}$ and $PM_{10}$) during January-December 2013 at a regional site NRCS (National
Reference Climatological Station) in Hangzhou, which is also an integrated measurement site for the research of climate
change and atmospheric environment. This study presents the first results of one-year measurements of trace gases and
particle matters in the urban area of Hangzhou, investigates the characteristics and cause of these chemicals by discussing
their seasonal characteristics, interspecies correlations, the concentration dependence on local emission and regional
transport, and the specific photochemical pollution and haze case, respectively.
**2 Introduction to the experiment, meteorological conditions, and methodology**
**2.1 Site description**
Hangzhou is situated in the eastern coast of China and is one of the most developed cities in the Yangtze River Delta region.
It has 8.9 million population and 2.7 million vehicles according to the 2014 Statistical Bulletin of Hangzhou. It belongs to
the subtropical monsoon climate, with an average temperature of 17.0°C, relative humidity of 75% and rainfall of 1438 mm
over the past 30 years (1981-2010). In this study, all in-situ measurements of gaseous constituents, particles and
meteorological factors were conducted at NRCS site (30.22°N, 120.17°E, 41.7 m a.s.l) in the center of Hangzhou (Fig. 1). As
the right top map shown in Fig. 1, the site is adjacent to Prince Bay Park (area, 0.8 km²) and situated in the northeastern of



West Lake famous scenic spot (area, 49 km$^2$) and commercial and residential areas in the south of city. There are no local
industrial pollution sources around the site. Thus, all gaseous constituents at this site can be representative of the urban areas
in Hangzhou.

**2.2 Measurements description**

Measurements of trace gases, aerosols, and meteorological parameters were conducted at NRCS station during January-
December 2013. Trace gases including $O_3$, $SO_2$, NO, $NO_y$, and CO were detected by a set of commercial trace gas analyzers
(Thermo Environmental Instruments Inc., USA i-series 49i, 43i, 42i, 42i-Y, and 48i), respectively, with a resolution of 1 min.
All the instruments are housed on the top floor of a laboratory building, which sits on the top of a hill about 40 m above the
ground level. Ambient air was drawn from the 1.5 m above the rooftop to the laboratory building through a manifold
connected to $O_3$, $SO_2$, NO and CO analyzers with PFA Teflon tubes (inside diameter: 2 cm). A separate sample line with a
MoO converter was used for $NO_y$ analyzer. All trace gas analyzers were weekly span and daily zero checked, and multi-
point calibration was made once a month.
Ambient $PM_{2.5}$ samples were collected using co-located Thermo Scientific (formerly R&P) Model 1405D samplers. The
sensor unit contains the two mass measurement hardware systems that monitor particles that continuously accumulate on the
system's exchangeable TEOM filters. PM-Coarse and $PM_{2.5}$ particulate, split by a virtual impactor, each accumulate on the
system's exchangeable TEOM filters. By maintaining a flow rate of 1.67 L·min$^{-1}$ through the coarse sample flow channel
and 3 L·min$^{-1}$ through the $PM_{2.5}$ sample channel, and measuring the total mass accumulated on each of the TEOM filters, the
device can calculate the mass concentration of both the $PM_{2.5}$ and PM Coarse sample streams in near real-time. TEOM filters
must be replaced before the filter loading percentage reaches 80% to ensure the quality of the data generated by the
instrument. For PM, the precisions of this instrument were 2.0 μg cm$^{-3}$ for 1 h average and 1.0 μg cm$^{-3}$ for 24 h average.

**2.3 Meteorological characteristic**

Table 1 shows the monthly averaged meteorological parameters at NRCS station, suggesting distinct characteristics of air
temperature in winter and summer in this region, with monthly averages from ca. 5 $^o$C in Janurary to ca. 32 $^o$C in July. High
relative humidity (RH) and a large amount of rainfall appeared in June (346 mm in total), and oppositely less precipitation
and low RH in autumn and winter. Note that the seemed high RH and large rainfall occurred in October was due to an
extremely synoptic event on 7 October, 2013 with the daily total rainfall of 91 mm. In addition, the wind rose implied that
the prevailing wind was from northwest in autumn, north in winter, and from southwest in spring and summer (See Fig. S1
in the Supplement).





**2.4 Methodology**
**2.4.1 Air mass back trajectory cluster**
In this study, 72-h back trajectories starting at the arrival level of 100 m from NRCS sites were calculated by using the
National Oceanic and Atmospheric Administration (NOAA) HYSPLIT-4 model with a $1^{o}\times1^{o}$ grid and the final
meteorological database. The six hourly final archive data were obtained from the National Center for Environmental
Prediction's Global Data Assimilation System (GDAS) wind field reanalysis. GDAS uses a spectral medium-range forecast
model. More details could be showed at http://www.arl.noaa.gov/ready/open/hysplit4.html. The model was run four times
per day at starting times of 00:00, 6:00, 12:00, and 18:00 UTC (08:00, 14:00, 20:00, and 02:00 LT, respectively). The
method used in trajectory clustering was based on the GIS-based software TrajStat (Wang et al. 2004).
**2.4.2 Potential source contribution function**
The potential source contribution function (PSCF) is widely used to identify regional sources based on the HYSPLIT model.
The zone of concern is divided into i×j small equal grid cells. The PSCF value in the ij-th cell is defined as mij/nij, where $n_{ij}$
is denoted as the numbers of endpoints that fall in the ij-th cell and $m_{ij}$ represents the numbers of "polluted" trajectory
endpoints in the ij-th cell. In this analysis, average concentrations were considered as the "polluted" threshold (Hsu et al.,
2003; Zhang et al., 2013). To minimize the effect of small values of $n_{ij}$, following the method of Polissar et al. (1999), the
seasonal PSCF values were multiplied by arbitrary seasonal weight functions $W_{ij}$, expressed by WPSCF, to better reflect the
uncertainty in the values for these cells. Geographic areas covered by more than 95% of the back trajectories are selected as
the study domain. In this study, our study domain was in the range of $15$-$55^{o}$N and $105$-$135^{o}$E. The resolution was $0.5^{o}\times0.5^{o}$.
$$W_{ij(spring)}=\begin{cases}1.00 & 36<n_{ij}\\0.70 & 12<n_{ij}\leq36\\0.42 & 6<n_{ij}\leq12\\0.17 & n_{ij}\leq6\end{cases} \quad W_{ij(summer)}=\begin{cases}1.00 & 42<n_{ij}\\0.70 & 14<n_{ij}\leq42\\0.42 & 7<n_{ij}\leq14\\0.17 & n_{ij}\leq7\end{cases}$$

$$W_{ij(autumn)}=\begin{cases}1.00 & 36<n_{ij}\\0.70 & 12<n_{ij}\leq36\\0.42 & 6<n_{ij}\leq12\\0.17 & n_{ij}\leq6\end{cases} \quad W_{ij(winter)}=\begin{cases}1.00 & 54<n_{ij}\\0.70 & 18<n_{ij}\leq54\\0.42 & 9<n_{ij}\leq18\\0.17 & n_{ij}\leq9\end{cases}$$

Moreover, to better elucidate the local and regional contribution to pollutants concentrations, we further compared the
WPSCF results with their corresponding emission inventories of $PM_{2.5}$, CO, $NO_x$, and $SO_2$ in 2013 provided by Peking
University (http://inventory.pku.edu.cn/), which were estimated by using a bottom-up approach with $0.1^{o}\times0.1^{o}$ spatial
resolution (Wang et al., 2013; Huang et al., 2014; Zhong et al., 2014), respectively.



### 2.4.3 Geopotential height (GH)

The geopotential height (GH) fields derived from the National Center for Environmental Prediction (NCEP) global Final (FNL) reanalysis (http://rda.ucar.edu/datasets/ds083.2/) are typically used to classify the synoptic types (Miao et al., 2017b). In this study, daily GH fields at the 925 hPa level from the NCEP-FNL reanalysis covering the region (100-135 $^o$E, 20-50 $^o$N) were classified to the prevailing synoptic types during photochemical pollution and haze episodes as discussed in Section 3.5. The NECP-FNL reanalysis was produced from the Global Data Assimilation System, which continuously assimilates observations from the Global Telecommunication System and other sources. The NECP-FNL reanalysis fields were on $1^o \times 1^o$ grids with a 6 h resolution.

## 3 Results and discussion

### 3.1 Concentration levels

To evaluate the overall concentration level of gaseous and particle pollution at NRCS, we selected a Grade II standard of the Chinese Ambient Air Quality Standards (CAAQS, GB 3095-2012), which was released in 2012 by the China State Council and implemented thorough the whole nation in 2016 (MEP, 2012). Inferred from the Grade II CAAQS for $PM_{2.5}$ (75 µg m$^{-3}$ for 24 h average) and $PM_{10}$ (150 µg m$^{-3}$ for 24 h average), 62 days and 26 days of $PM_{2.5}$ and $PM_{10}$ exceedances with daily average of 102.2 µg m$^{-3}$ and 195.3 µg m$^{-3}$ were classified thorough the period, respectively, mostly occurred in winter. For $O_3$, about 38 days exceedances (93 ppbv for 1 h average for the Grade II CAAQS) in whole were found during the whole period, mostly covering from May to September. It suggested Hangzhou was suffering from heavy haze and photochemical pollution in cold and warm seasons. Concerning $SO_2$, the annual mean was 10.9 ppbv in this study, nearly half of the yearly mean of $SO_2$ Grade II CAAQS (21 ppbv). It was reasonably attributed to the powerful measure of Chinese government to control the emission of $SO_2$ starting at 1990 (He et al., 2002; Qi et al., 2012). Table 2 summarized a statistical analysis on these species and listed the comparison with the previous results in other typical regions in China. In general, with respect to all these chemicals, our results were generally comparable with those observed by other contemporaneous measurements in Hangzhou and the other cities in YRD. As expected, regional differences between YRD, PRD, and BTH could be also found as illustrated in Table 2. For instance, observed $PM_{2.5}$, $PM_{10}$, and CO concentrations were higher in BTH than those in YRD and PRD through the comparison among provincial capital cities in China during 2011-2014 (Chai et al., 2014; Wang et al., 2014), which has been extrapolated to be more emissions from coal-based industries and coal and biomass burning based domestic home heating in BTH in winter (Zhang et al., 2012; Yang et al., 2013; Chai et al., 2014). Moreover, slight decreases in $PM_{2.5}$ and $PM_{10}$ at NRCS were both evidenced by their respective difference between 2013 and 2010-2011 (Tab. 2), coincident with the results derived from the satellite data and ground monitoring in China (Ma et al., 2016; Seltenrich, 2016). For $NO_y$, only rough comparison was implemented due to very limited measurements executed in China. The yearly mean $NO_y$ concentration of 63.7 ppbv in this study was slightly higher than 54.6 ppbv in Beijing (Wu et al., 2016b). It's interestingly noted that slightly higher $NO_y$ at NRCS possibly indicated more photochemical conversion of $NO_x$ in



Hangzhou than Nanjing in the presence of nearly identical $NO_2$. Additionally, the daytime mean concentrations were
comparable with those at nighttime for $PM_{2.5}$ nearly in all seasons but higher for $O_3$ due to the daily variations in solar
radiation and air temperature, the reverse is true for CO, $NO_x$, and $NO_y$.

## 3.2 Seasonal characteristics

Figure 2 shows seasonal variations of atmospheric $O_3$ (a), CO (b), NO (c), $NO_x$ (d), $NO_y$ (e), $O_x$ (f), $PM_{2.5}$ (g), $PM_{10}$ (h), and
$SO_2$ (i). Ozone exhibits a distinguished seasonal variation, with a board peak in late spring and middle summer (a maximum
in May and a secondary maximum in July) and a minimum in winter (November to January).Its observed behavior at NRCS
is different from what has been disclosed in previous studies conducted in southern and northern China, such as a summer
minimum and an autumn maximum of $O_3$ found in Hong Kong and an early summer (June) broad maximum recorded in
Beijing (Ding et al., 2008; Lin et al., 2008, 2009; Zhang et al., 2014). Recently, Ding et al. (2013) presented two peaks of $O_3$
appearing in summer (July) and early autumn (September) at Xinlin site in the suburban area northeast of Nanjing (about
239 km away from NRCS station). Regarding the geographical location of Hangzhou, which is upwind of the YRD under
the influence of southeasterly summer monsoon, the emissions in the YRD region and the solar radiation might be the main
causes of an $O_3$ formation in summer, resulting in a different seasonal cycle of $O_3$ compared to other continent sites in the
west/northwest YRD. In fact, the CO and $NO_y$ data (Fig. 2b and Fig. 2e) show that these precursors were still at fairly high
levels (about 500 ppbv and 35 ppbv, respectively) in summer. The low $O_3$ level in winter, especially at night, can be
attributed to the lower temperature, weaker solar radiation, and in particular the strong destruction of $O_3$ by chemical titration
of NO from local emission or regional transport as discussed below (Lin et al., 2008, 2009, 2011). Note that, a slight drop of
$O_3$ was found in June compared with other months in summer, mainly attributing to the more frequent rainy days (23 days)
and larger rainfall in June (346 mm) than those in May (15 days) and July (5 days) during summertime (Table 1).
For $PM_{2.5}$ and $PM_{10}$, Fig. 2g and -2h both displayed overall well-defined seasonal variations with the maximum in winter
(December) and the minimum in summer (July). In cold seasons the emission of particulate matter is normally high due to
more emission of fossil fuels during heating in northern China (Zhang et al., 2009), which contributed to the enhancements
of particle matters and other tracer gases (i.e., CO and $NO_x$) at NRCS site via long-distance transport (See discussion in
Section 3.4). Furthermore, in winter temperature inversion and low mixing layer contribute to decrease particle suspension
and advection (Miao et al., 2015a). Also, dry/wet deposition should have strong seasonal variations because high
precipitation favors wet-deposition and high soil humidity, and the growth of deciduous plants may also favor the dry
deposition of particle matter in warm seasons (Zhang et al., 2001). The relatively low concentrations of $PM_{2.5}$ and $PM_{10}$ in
summer may be also partly due to an increased vertical mixing (i.e., a higher boundary layer height) and more convection
(Ding et al., 2013; Miao et al., 2015b). $PM_{2.5}$ mass concentration also show strong month-to-month variations. The
simultaneous drop of $PM_{2.5}$ and $PM_{10}$ concentrations together with other primary pollutants (i.e., $SO_2$, CO and $NO_y$) in
February was mainly ascribed to the winter break of the Chinese Spring Festival, which started at the end of January and



lasted until mid-February. Notably, the seasonal pattern for PM was similar to $NO_x$, which suggested that traffic and heating
emissions were important to the $PM_{2.5}$ variation.
For other trace gases (CO, $NO_x$, $NO_y$, and $SO_2$), they all revealed clear seasonal variations but also some unique month-to-
month variation patterns (Fig. 2a-2f and Fig. 2i). Similar seasonal patterns among CO, $NO_x$, and $SO_2$ were generally found
with pronounced minimums appearing in summer and higher levels in fall and winter. Similar reasons with particle matters
could interpret these seasonal patterns such as the variation in the boundary layer height and the long-distance transport as
mentioned above. The last but not the least was photochemistry. During summer, it's most active to accelerate the
transformation of primary gaseous pollutants, whereas in winter, weaker photochemical reaction cannot remove the gases as
quickly as in the warmer seasons from the atmosphere.
$NO_y$ concentration increased at the end of autumn, with a maximum in December together with a sharp peak of NO. Time
series implied that in December there was a multi-day episode of $NO_x$ with high mixing ratios of NO and $NO_2$ both reaching
up to 100 ppbv and these days were generally correlated with northwest wind, suggesting a fresh emission from factories in
the industrial zone in the northwest. The "potential ozone" $O_x$ ($O_3$ +$NO_2$) is usually used as an estimate of atmospheric total
oxidant (Lin et al., 2008). In summer (Fig. 2f), an abnormally high level of $O_x$ (mainly as $NO_2$) was found in winter but for
$O_3$ decreased. The high level of $NO_2$ in $O_x$ was expected to be originated from the significant titration of high NO by $O_3$ in
November and December (Fig. 2a).
As shown in Fig. 2i, $SO_2$ displayed a strong increase in winter but a significant drop in November. This pronounced
winter peaks were mainly due to the increased coal consumption for heating as mentioned above. The drop was associated
with the $PM_{2.5}$ maximum and a relatively high RH (Fig. 2g and Table 1), suggesting a possible role of heterogeneous
reactions (Ravishankara, 1997).
**3.3 Inter-species correlations**
Inter-species correlation could be normally used as an agent for acquiring some insights on their chemical formation,
removal processes, and interactions. As displayed in Fig. 3-7, we presented scatter plots of $O_3$-$NO_y$, $PM_{2.5}$-$NO_y$, $SO_2$-$NO_y$,
CO-$NO_y$, and $PM_{2.5}$-$O_3$ correlations based on the whole dataset, respectively, and further discriminated these correlations
under typical environmental or meteorological impacts with color-coded parameters (i.e., relative humidity, air temperature,
and $O_3$ concentration). Clearly, overall negative correlation was found between $O_3$ and $NO_y$ during the whole period (Fig. 3).
The color data showed that negative correlation mainly appeared with data of low air temperature, implying a remarkable
titration of freshly emitted NO with $O_3$ during the cold seasons and at nighttime. In contrast, a positive correlation between
$O_3$ and $NO_y$ dominated under high air temperature, which usually occurred in the daytime of warm seasons within a
moderate level of $NO_y$ (<150 ppbv). These findings suggested a strong local photochemical production of $O_3$ in summer,
leading to its seasonal variations as illustrated in Fig. 2a.
As illustrated in Fig. 4, a good positive correlation was found between $PM_{2.5}$ and $NO_y$, suggesting that $PM_{2.5}$ was highly
correlated with fossil combustion at this site. Some green data in the plot show very high $NO_y$ concentration together with





low PM$_{2.5}$, suggesting that high NO air masses during December. Fig. 4 exhibited that high RH data were very scattered but
not very high PM$_{2.5}$/NO$_y$, implying that negligible interference of humidity on TEOM PM$_{2.5}$ measurement during the study
period, even under high RH condition in summer.
SO$_2$ and NO$_y$ show a moderate to good correlation (See Fig. 5). Specifically, a better correlation and higher SO$_2$/NO$_y$ ratio
were gained from air with low humidity. Nevertheless, the point distribution was much more scattered for the humid air
masses, and the ratio of SO$_2$/NO$_y$ was clearly low, confirming a higher conversion of SO$_2$ to sulfate and/or deposition in the
humid condition (Khoder, 2002; Su et al., 2011). In this study, the averaged ratios of SO$_2$/NO$_y$ during 18 February-30 April
was lower as 0.017, compared with that previously reported at Lin'an during the same months twelve years ago (Wang et al.,
2004). It was mainly owing to a remarkable reduction of SO$_2$ emission from power plants but an increased NO$_x$ emission
associated with a huge consumption of petroleum fuels in the past decade in this region (Zhang et al., 2009).
Figure 6 shows a good positive correlation between CO and NO$_y$ coded with O$_3$ mixing ratios. For CO lower than 3.2
ppmv during the whole period, an increase of NO$_y$ generally leaded to lower O$_3$ concentrations, but CO reversed. As a
common origin of VOCs and CO, VOCs play a similar behavior with CO in the ozone photochemistry. Our results suggested
a VOCs-limited regime throughout the year in Hangzhou, consistent with the reported results in other cities of YRD region
(e.g., Shanghai and Nanjing) (Geng et al., 2007; Ding et al., 2013). While, as specifically shown in Fig. 6b, atmospheric O$_3$
(above 80 ppbv), mainly occurred in the afternoon (14:00-16:00 LT) in the summer and early autumn, exhibited increased
trend with the increasing NO$_y$ within air masses with moderated CO mixing ratio of 0.25-1.5 ppmv, and the reversed trend
for CO was not expected to be significantly increased. It indicated that the transition from VOCs-limited regime to an
optimum O$_3$ production zone (even NO$_x$-limited regime), probably occurred at NRCS site in warmer seasons. We speculated
this change was mainly attributed to the larger emission of biogenic VOCs (BVOCs) compared to cold seasons. As reviewed
by Calfapietra et al. (2013), the VOC-limited conditions, in which O$_3$ production is limited by a high concentration of NO$_x$,
are often observed in urban areas. However, if high BVOC emitters are common in urban areas, they could move the
VOC/NO$_x$ ratio toward optimal values for O$_3$ formation, and resulted in this ratio reaching in the city centers. As depicted in
Section 2.1, our study site is situated adjacent to Prince Bay Park (area, 0.8 km$^2$) and in the northeastern of West Lake
famous scenic spot (area, 49 km$^2$). For these two regions, they were both block green parks with high vegetation coverage.
Moreover, the primary tree species in these two regions are Liquidambar formosana and Cinnamomum camphora,
respectively, as major contributor to the emissions of isoprene and monoterpene (Chang et al., 2012), favoring the formation
of O$_3$. Air masses from Prince Bay Park and West Lake famous scenic spot were confirmed to be transported to NRCS site
during warmer seasons, as illustrated in Fig. S1 and Fig. 8b. In addition to the strong temperature dependence of isoprenoid
emission (Guenther et al., 1995), a significantly increased emission of BVOCs was expected and thus it disturbed the
original balance between VOCs and NO$_x$ relative to cold seasons. Our conclusion was generally in line with the
contemporaneous study implemented by Li et al. (2016a) who found that VOCs-limited regime accounted for 47% of the
ozone formation during the summer in Hangzhou, and the others are under NO$_x$-limited, taking BVOCs into consideration.



Recently, Li et al. (2017) also deduced the summer ozone mostly presented VOCs-limited and transition region alternately in
urban Hangzhou.
A scatter plot of $O_3$ with $PM_{2.5}$ color-coded with air temperature was depicted in Fig. 7. During moderate to high air
temperature, a significant positive correlation was elucidated between $O_3$ and $PM_{2.5}$ and the reverse negative correlation was
found under low temperature. The positive correlation for warm air probably reflected a formation of secondary fine
particles in summer associated with high $O_3$. The secondary particle formation may be related to high conversion rate of $SO_2$
to sulfate under a high concentration of oxidants (Khoder, 2002). Additionally, it was also associated with the formation of
secondary organic aerosols with high $O_3$ concentrations (Kamens et al., 1999; Lambe et al., 2015; Palm et al., 2017), which
was primarily produced through the photo-oxidation of BVOCs (Claeys, et al., 2004; Böge et al., 2013). As inferred above,
significant emission of BVOCs was speculated around NRCS in summer. The anti-correlation for cold air might be caused
by the titration effect of high NO concentration, which was in relation to high primary $PM_{2.5}$ in cold seasons.
**3.4 Dependences of pollutant concentrations on local emission and regional transport**
To overview the impact of wind on the pollutants concentrations, we draw the seasonal wind dependence maps of pollutants
concentrations with wind sectors (See Fig. S2 in the Supplement for details). In total, similar seasonal patterns of wind
dependence map were found between CO and $PM_{2.5}$, $SO_2$, and $NO_y$ ($NO_x$), in good agreement with their seasonal patterns as
shown in Section 3.2. For CO and $PM_{2.5}$, their top 10% concentrations were generally related with all the directions
throughout the year at speeds lower than 2 m s$^{-1}$ while bottom 10% were associated with others direction wind except north
at higher wind speed. It's necessary to pay attention on the scatter points of top 10% concentrations distributed in north
direction with high wind speed. With respect to the wind direction and transport, as the wind speed increases, pollutants
concentrations should have been decreasing due to the more effective local dilution, thus the increase instead might indicate
potential sources in these directions.
To address this issue and further investigate the relative contribution of local emission and regional transport, we
employed the trajectory clustering and WPSCF, along the comparison with the emission inventories. The 72 h back
trajectories from NRCS site were computed by using HYSPLIT model for four seasons. As shown in Fig. 9a, we obtained
six clusters by the clustering algorithm for four seasons with seven dominant paths distributed in east (E), northeast (NE),
north (N), northwest (NW), west (W), southwest (SW), and southeast (SE). The length of the cluster-mean trajectories
indicates the transport speed of air masses. In this analysis, the long and fast moving trajectories were disaggregated into
groups originating from more distant SE and SW regions during summer and NW and N regions during other seasons.
Member of this cluster have extremely long transport patterns, some of them even cross over Inner Mongolia and Mongolia
(e.g., N and NW). Trajectories belonging to S-SW and E-SE typically followed flow patterns from South Sea and Pacific
Ocean, respectively. Otherwise, some trajectories have short transport patterns, indicative of slow-moving air masses. Most
of the pollution episodes within this group are probably enriched from regional and local emission sources. Such trajectories
were also identified during every season in our study. For instance, the air masses associated with cluster 4 (in spring,



autumn, and winter) and cluster 1 in summer were predominantly originating from local areas and the nearby provinces with
significant pollution sources such as Jiangsu, Anhui, and Shanghai.
Table 3 summarizes the percentages of these identified trajectory clusters on seasonal basis and the corresponding mean
concentrations of $PM_{2.5}$ and other trace gases related to each trajectory cluster. As inferred from Table 3, the clusters
exhibited larger variability and season dependence: the predominant clusters were W (42.66%) in spring, SW (53.89%) in
summer, NW (35.53%) in autumn, and N (54.91%) in winter, respectively. It's of interest to note that some trajectory
clusters with small percentages are remarkably related with high pollutants concentrations. In summer, a few $PM_{2.5}$ pollution
cases (only 8.42% of the summertime trajectories) with mean concentration as high as 51.5 µg m$^{-3}$ were related with the N
trajectories travelling across well-industrialized cities cluster (i.e., Suzhou, Wuxi, and Changzhou).
Furthermore, we depicted the seasonal WPSCF maps (a), the corresponding zoomed maps (b), and the emissions maps (c)
for $PM_{2.5}$, $O_3$, CO, $NO_x$, and $SO_2$, respectively, denoted with alphabets a, b, and c in the figure captions. Here we presented
the results of two representative species $PM_{2.5}$ (Fig. 9a, -9b, and -9c) and $O_3$ (Fig. 10a, -10b) and those of the other species
were included in the Supplement (Fig. S3a, -S5c). Judging from the WPSCF maps, together with their corresponding
zoomed views and the calculated emissions maps, a few distinct features were summarized: (1) Local emissions were both
significant for the primary pollutants such as CO (Fig. S3), $NO_x$ (Fig. S4), $SO_2$ (Fig. S5), and $PM_{2.5}$ (Fig. 9) on seasonal scale.
For $O_3$, local photochemistry dominated during spring, summer, and autumn (Fig. 10a, -10b) due to strong photochemical
reactivity; (2) long transports from Yellow Sea, East Sea, and South Sea were also important potential sources for $NO_x$ (Fig.
S4a) and $O_3$ (Fig. 10a, -10b); (3) The potential sources of CO and $NO_x$ had similar patterns on spatial and seasonal scales,
with higher values in the NW during spring, covering the mid-YRD regions across Anhui Province and reaching the border
of Henan Province; in the NW and N during autumn and winter, covering the most area of Jiangsu Province and part of
Shandong Province such as Jinan, and Zibo city; (4) the higher values for $SO_2$ were located in the Ningbo city and the coast
of Yellow Sea during spring, in the southeastern region from East Sea during summer, probably due to ship emissions (Fan
et al., 2016), but in the inland cities such as Shaoxing and Quzhou city of Zhejiang Province during autumn and Anhui
Province during winter. In total, along with the air mass trajectories, the WPSCF maps for these primary pollutants were
generally in line with their respective corresponding species' emissions (Fig. 9c, -S3c, -S4c, and –S5c). Although no
seasonal patterns in emission maps were found, the emissions of these pollutants exhibited interspecies similarity and strong
spatial dependence with industrialization level. Note that the emission of $NO_x$ was significant from South Korea (Fig. S4c)
where high WPSCF values were found in autumn (Fig. 10a), indicating a remarkable source to the surface $O_3$ of NRCS
through the northeasterly transport.
In terms of $PM_{2.5}$, the potential sources showed distinct seasonal variations such as southeastern regions of Jiangxi
Province and northwestern area of Zhejiang Province during spring and in the western city of North Korea (Pyongyang) and
South Korea (Seoul) with the northeasterly air mass across Yellow Sea during summer. As illustrated in Fig. 9a and -9b, the
contributions from local emission were both found to be more significant for autumn and winter than spring and summer,
covering all the cities in Zhejiang Province especially for the southern and southwestern part (e.g., Lishui, Jinhua, and





Quzhou city). Moreover, we found the higher WPSCF values located in the middle city of Jiangsu Province in autumn and
the expanded area towards the whole Jiangsu and Anhui Province and the southeast coast cities (e.g., Wenzhou, Ningbo in
Zhejiang Province, Fuzhou in Fujian Province) in winter, revealing the cross-boundary transport is crucial to the pollution of
particle matters. This result has been confirmed by Yu et al. (2014) who also found such transport dominated in the
Hangzhou city during the heavy haze episode (3-9 December, 2013).

For $O_3$, its potential sources exhibited distinct seasonal and spatial distributions: apart from the local contribution as

discussed above, the results with high WPSC values, as illustrated in Fig. 10a, indicated the main potential sources were
located in the western and southwestern region (e.g., Anhui, Jiangxi, and mid-Guangdong Province), and the northwestern
area such as Jiangsu, Henan, and Shandong Province during spring; In summer, more extensive potential sources were
elucidated to be located in the eastern-southern-southwestern regions of China, covering the southern part of Zhejiang
Province, southeastern cities of Jiangxi Province, almost the whole Fujian Province, and the eastern part of Guangdong
Province; the mid-Zhejiang Province (e.g., Quzhou, jinhua, and Ningbo city) and the northern coastal cities (e.g., Shanghai,
Lianyungang, and Dalian city) were apparently potential sources in autumn; in regard to winter, long distant transport acted
as a significant source of surface $O_3$, specifically from the northeasterly air mass Yellow Sea. A very interesting finding
should be pointed out that offshore area of East China Sea, Yellow China Sea, or even far from South China, respectively on
southeastern Zhejiang, Jiangsu, and Fujian Province were significant sources of $O_3$ at NRCS site throughout the year. We
speculated the recirculation of pollutants by sea- and land-breeze circulations around the cities along the YRD and Hangzhou
Bay which has been confirmed by Li et al. (2015, 2016b), was largely responsible for the increased concentration of $O_3$ at
NRCS site. Such an increase in $O_3$ concentrations in urbanized coastal areas have been observed and modeled in a number of
studies (Oh et al., 2006; Levy et al., 2008; Martins et al., 2012). Thus, our study further emphasizes the importance of local
thermally induced circulation on air quality.
**3.5 Cases studies for haze (high PM$_{2.5}$) and photochemical pollution (high O$_3$) episodes**
To elucidate the specific causes of high PM$_{2.5}$ and $O_3$ episodes including the transport and local photochemical formation, we
chose two typical cases for detailed interpretations and are presented here. In this study, the haze pollution episode is defined
as the event that continuous days with daily averaged PM$_{2.5}$ concentration exceeding 75 μg m$^{-3}$, which has been also used to
distinguish non-haze and haze episode in other studies (Yu et al., 2014; Wu et al., 2016a). With respect to this campaign,
there were two non-haze episodes (Phase I (28 Nov.-3 Dec.), Ⅱ (10-12 Dec.)), and their subsequent severe haze pollution
episodes (Phase Ⅲ (2-9 Dec.) and Ⅳ (13-15 Dec.)) at NRCS site, respectively, as illustrated in Fig. 11. In the Phase Ⅲ, it
showed that high PM$_{2.5}$ (up to 406 μg m$^{-3}$) appeared on 7 December and board PM$_{2.5}$ peaks (around 300 μg m$^{-3}$) occurred
before and after two days. Simultaneously, CO, SO$_2$, and NO$_x$ also reached very high levels on this day, confirming that the
common origin of CO and PM$_{2.5}$ from heating and combustion and the rapid conversion of SO$_2$ and NO$_x$ to sulfate and
nitrate in PM$_{2.5}$ in winter. But for $O_3$, its level reached as low as 11.5 ppbv at 15:00 LT on that day, owing to the weak
photochemical activity under the severe haze pollution. Along with the high NO$_2$ concentration (around 120 ppbv), it could



not produce sufficient conversion oxidants (OH and $HO_2$ radicals) for the gas-phase oxidation of $SO_2$ (Poppe at al., 1993;
Hua et al., 2008), while the increased relative humidity during 6-8 December possibly favored the aqueous phase oxidation
of $SO_2$.
Moreover, according to the results obtained from the backward trajectory cluster and WPSCF analysis during 2-9
December, 2013 (Fig. S6 in the Supplement), we found an apparent contribution from the transported air mass from
northwest region such as Jiangsu Province and Anhui Province. Our results were in good agreement with contemporaneous
measurement in Hangzhou (Wu et al., 2016a). Subsequently, at the end of this episode significant drops of these species
except $O_3$ were observed from 00:00 LT to 23:00 LT on 9 December (i.e., 189 to 41.6 µg m$^{-3}$ for $PM_{2.5}$, 2.3 to 1.0 ppmv for
CO, and 145 to 47.9 ppbv for $NO_x$). Weather chart and wind data suggested that the region of NRCS was always controlled
by a strong continental high pressure system originating from northwest before 8 December (Fig. 12a-12f), but rapidly
changed to be dominated under a strong marine high pressure system coming from east at 02:00 LT on 9 December (Fig.
12g-12h), which brought clean maritime air passing over Yellow Sea and thus caused such decreases in these pollutants.
However, it quickly turned back to be controlled under a continental high pressure system described above, carrying
pollutants from the city clusters to the NRCS site. It could account for the accumulations of these species during the
intermediate period (Phase Π). For the subsequent Phase IV with high $PM_{2.5}$ episode it was also found to be governed by a
stagnant high pressure over YRD region (Fig. S7).
For the photochemical pollution events, we selected three cases with $O_3$ exceedances during May-August according to
Grade II standard of CAAQS as depicted in Section 3.1. As displayed in Fig. 13, they were the Phase I (28-30 May and 20-
22 June) with rapid buildup and decrease of $O_3$ within 3 days, Phase Π (9-12 July) representing a distinct accumulation
process of $O_3$ exceedances, and the Phase Ш (1-3 May, 20-22 May, and 9-11 August) with high $O_3$ levels within three
consecutive days. For 28 May in the Phase I, weather chart suggested that a strong anticlockwise cyclone located over YRD.
In this case, the cyclone (i.e., low pressure) caused favoring conditions, e.g., cloudy weather and high wind velocities, for
pollution diffusion. Then, a strong clockwise anticyclone from northwest, sweeping over cities cluster (i.e., Nanjing and
Shanghai), rapidly moved adjacent to NRCS site on 29 May. It carried the primary pollutants such as CO, $SO_2$, $NO_x$ from
these megacities and secondary products (i.e., $O_3$ and some $NO_z$) were further produced via complex photochemical
reactions under such synoptic conditions. As orange shaded area shown in Fig. 13, the hourly maximums of $O_3$ and $PM_{2.5}$
were observed as high as 141.2 ppbv and 135.8 µg m$^{-3}$ at 13:00 LT on 29 May. Following this day, the cyclone again
dominated this region and caused sudden decreased in atmospheric pollutants. Also, similar case was found during 20-22
June under such changes in synoptic weather. For Phase Π (9-12 July), a typical accumulation process was observed with the
daily maximums of atmospheric pollutants increasing from 90.4 to 142.9 ppbv for $O_3$, 77.6 to 95.3 µg m$^{-3}$ for $PM_{2.5}$, and
80.2 to 125.2 ppbv for $NO_y$, respectively. The examination of day-to-day 925-hPa synoptic chart derived from NECP
reanalysis suggested that high pressure system governed over YRD during 9-11 July, with southwesterly prevailing wind.
The air masses recorded at this site mainly came from the most polluted city clusters in the southwest (e.g. Zhejiang, Jiangxi,
and Fujian Province). Meanwhile, the stagnant synoptic condition (i.e., low wind speed) favored the accumulation of



primary pollutants such as CO and $NO_x$. For secondary pollutants $O_3$ and $PM_{2.5}$, they were also rapidly formed via
photochemical oxidation and further accumulated under such synoptic condition, together with continuous high-temperature
(daily mean around 33 $^o$C). On 12 July, a typhoon (No. 7 Typhoon Soulik) moved to a location a few hundred kilometers
away from NRCS site, bringing southeasterly maritime air over YRD. Daily maximum $O_3$ reached at 142.8 ppbv at 12:00 LT
even with low concentration of precursors (i.e., 0.48 ppmv for CO and 16.0 ppbv for $NO_x$), suggesting high photochemical
production efficiency of $O_3$ in this region in summer. This phenomenon has been also found in the multi-day episode of high
$O_3$ in Nanjing during 20-21 July, 2011 (Ding et al., 2013). In this phase, $PM_{2.5}$ mass concentration showed very good
correlation (R = 0.79, p < 0.001) with $O_3$ during the daytime (09:00-17:00 LT), possibly indicating a common origin of
BVOCs due to the significant vegetation emission as discussed above, in addition to high biomass production in the southern
part of the YRD (Ding et al., 2013). For Phase Ш (1-3 May, 20-22 May, and 9-11 August), there were most sunny days with
low wind speed and moderate/high air temperature which were both beneficial factors for photochemical formation of $O_3$,
together with sufficient precursors ($NO_x$ and VOCs) in the summer and early autumn over YRD. For 1-3 May and 20-22
May, daily maximum T were moderate (around 25 $^o$C versus 31 $^o$C), while the daily maximums $NO_x$ reached as high as 43-
95 ppbv and 50-90 ppbv, respectively, which both favoring the photochemical formation to produce the continuous high $O_3$
concentrations (daily maximums: 96-133 ppbv via 104-133 ppbv). The reverse case is also true during 9-11 August, on
which the daily maximum T and $NO_x$ ranged from 40.6-41.4 $^o$C and 33-44 ppbv, respectively, resulting in producing
continuously high $O_3$ from 98.8 ppbv to 130.5 ppbv.

**3.6 Photochemical age and ozone production efficiency during photochemical pollution and haze period**

Photochemical age is often used to express the extent of photochemistry, which can be estimated using some indicator such
as $NO_x/NO_y$ (Carpenter et al., 2000; Lin et al., 2008, 2009, 2011; Parrish et al., 1992). Air masses with fresh emissions have
$NO_x/NO_y$ close to 1, while lower $NO_x/NO_y$ ratio for the photochemical aged air masses. In this study, for the haze events as
mentioned above, the average and maximum $NO_x/NO_y$ ratios were as high as 0.80 and 0.99, respectively, indicating that
photochemical conversion of $NO_x$ is not absent but fairly slow. It was well consistent with the largely weaken
photochemistry due to the low intensity of UV radiation in winter. In contrast, during the photochemical pollution period,
they were low as 0.53 and 0.14 for the average and minimum ratio. The simultaneous measurements of atmospheric $O_3$, $NO_x$,
and $NO_y$ can provide an insight into calculating the ozone production efficiency (OPE) for different seasons. From the data
of $O_x$ and $NO_z$, the ratio of $\Delta(O_x)/\Delta(NO_z)$ can be calculated as a kind of observation-based OPE (Trainer et al., 1993; Sillman,
2000; Kleinman et al., 2002; Lin et al., 2011;). In this study, the mean values of $NO_z$ and $O_x$ between 07:00-15:00 LT, were
used to calculate the OPE values through the linear regressions. In addition, these data were also confined to the sunny days
and the wind speed below 3 m s$^{-1}$, reflecting the local photochemistry as possible. The OPE value during the photochemical
pollution period (SOPE) as mentioned above was 1.99, generally within the reported range of 1-5 in the PRD cities, but
lower than 3.9-9.7 in summer Beijing (Chou et al., 2009; Ge et al., 2012). Meanwhile, the OPE value of 0.77 during the haze
period (HPOE) was also comparable with the reported value of 1.1 in winter in Beijing (Lin et al., 2011). The smaller winter



OPE value in Hangzhou might be ascribed to the weaker photochemistry and higher $NO_x$ concentration. At high $NO_x$ level, OPE tends to decrease with the increased $NO_x$ concentration (Ge et al., 2010; Lin et al., 2011). In Hangzhou, the $NO_x$ level is frequently higher than needed for producing photochemical $O_3$, and excessive $NO_x$ causes net $O_3$ loss rather than accumulation. In this study, 75% of daily OPE values were negative, for which two factors could accounted. To some extent, due to the geographical location and unique climate characteristic for Hangzhou as depicted above, the interference of unbeneficial meteorological condition existed in the formation of local $O_3$ deriving from photochemistry, i.e., strong wind, frequent rainy days. The other one is because of the consumption of $O_3$ by excessive $NO_x$, which was also well confirmed by the conclusion that Hangzhou was mostly in the VOCs-limited regime as discussed in Section 3.2. Such circumstance was also observed at the rural site Gucheng in the NCP and in Beijing urban area (Lin et al., 2009, 2011). Taking the average of SOPE of 1.99 and the average daytime increment of $NO_z$ (ca. 20 ppbv), we estimated an average photochemical $O_3$ production of about 39.8 ppbv during photochemical pollution period. In contrast, the lower average photochemical $O_3$ production was estimated to be 10.78 ppbv during haze period based on HOPE, which might act as a significant source for surface $O_3$ in winter in Hangzhou.

## 4 Conclusions

In this study, we presented an overview of one year measurements of trace gases ($O_3$, CO, $NO_x$, $NO_y$, and $SO_2$) and particle matters ($PM_{2.5}$ and $PM_{10}$) at National Reference Climatological Station in Hangzhou. The characteristics and cause of these chemicals were investigated by their seasonal characteristics, along the comparison with the previous results in other regions in China, interspecies correlations, and the concentration dependence on local emission and regional transport. Specific photochemical pollution and haze case were studied in detail based on discussing the physical process and photochemical formation (ozone production efficiency). The main findings and conclusions are summarized below:

a) Within one year study period, there were 38 days of $O_3$ exceedances and 62 days of $PM_{2.5}$ exceedances of the National Ambient Air Quality Standards in China at the site, suggesting heavy air pollution in this region. In general, the concentration levels of these chemicals were consistent with those observed by other contemporaneous measurements in Hangzhou and the other cities in YRD, but lower than those in NCP. Distinct seasonal characteristics were found with a board peak in late spring and middle summer and a minimum in winter for $O_3$, while with maximum in winter and minimum in summer for $PM_{2.5}$.

b) A positive $O_3$-$NO_y$ correlation was found for air masses with high air temperature in summer, suggesting a strong local photochemical production of $O_3$. In addition, correlation analysis shows an important conversion of $SO_2$ to sulfate and/or deposition in the humid condition. CO-$NO_y$-$O_3$ correlation suggested a VOC-limited regime for the overall study period but moved toward an optimum $O_3$ production zone during warm seasons. The postive correlation between $O_3$ and $PM_{2.5}$ under high air temperature indicated a formation of secondary fine particles in warm seasons, respectively.

c) The results from the emission inventories of the primary pollutants such as $PM_{2.5}$, CO, $NO_x$, and $SO_2$ demonstrated that local emissions were both significant for these species but without distinct seasonal variations. The major potential sources



of $PM_{2.5}$ were located in the regions of southwesterly in spring, northwesterly and northeasterly in summer, and northwesterly (the whole Jiangsu Province and Anhui Province) in autumn and winter, respectively. For CO and $NO_x$, they showed similar patterns with northwestern regions covering the mid-YRD regions and Anhui Province during spring and in the northwestern and northern regions including Jiangsu Province and part of Shandong Province during autumn and winter. The distinct seasonal variation in $SO_2$ potential might be from southwestern and eastern region during spring and summer but northwestern during autumn and winter. Air masses transported from Yellow Sea, East Sea, and South Sea were also important in increasing surface $O_3$, probably due to the recirculation of pollutants by sea- and land-breeze circulations around the cities along the YRD and Hangzhou Bay. This finding further emphasizes the importance of urban-induced circulation on air quality.

d) Case studies for photochemical pollution and haze episodes both suggest the combined importance of local atmospheric photochemistry and synoptic weather during the accumulation (related with anticyclones) and dilution process (related with cyclones) of these episodes. The average photochemical $O_3$ productions were estimated to be 39.8 and 10.78 ppbv during photochemical pollution and haze period, respectively, indicating local photochemistry might act as a significant source for surface $O_3$ in winter in Hangzhou.

Our study further completes a picture of air pollution in the YRD, interprets the physical and photochemical processes during haze and photochemical pollution episodes, and explores the seasonal and spatial variations in the potential sources of these pollutants. Moreover, this work suggests the cross-region control measures are crucial to improve air quality in the YRD region, and further emphasizes the importance of local thermally induced circulation on air quality.

*Acknowledgement.* This study is financially supported by National Natural Science Foundation of China (41505108 and 41775127), National Key Research and Development Program of China (2016YFC0202300), and Shanghai Key Laboratory of Meteorology and Health (QXJK201501). The authors are especially grateful to Dr. Miao Yucong for the technical supports in drawing a part of figures and discussions.

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





Table 1 Statistics of general meteorological parameters at NRCS for the period during January- December 2013*.

| Month | Temperature (°C) | RH (%) | Wind Speed (m s⁻¹) | Rainfall (mm) | Pressure (Pa) | Visibility (m) |
|---|---|---|---|---|---|---|
| Jan. | 4.5 | 76 | 1.9 | 24.9 | 10221.6 | 2566.0 |
| Feb. | 7.0 | 81 | 2.3 | 66.8 | 10197.2 | 3511.8 |
| Mar. | 12.3 | 67 | 2.7 | 115.9 | 10140.5 | 5459.1 |
| Apr. | 16.9 | 56 | 2.6 | 98.1 | 10095.3 | 7587.8 |
| May | 23 | 69 | 2.1 | 121.3 | 10045.8 | 6118.9 |
| Jun. | 24.7 | 78 | 2.0 | 346 | 10013.0 | 5693.5 |
| Jul. | 32.2 | 51 | 2.8 | 9.3 | 9997.8 | 17011.0 |
| Aug. | 31.3 | 58 | 2.6 | 212.1 | 10001.7 | 13958.3 |
| Sep. | 24.5 | 73 | 2.3 | 49.4 | 10015.2 | 9584.7 |
| Oct. | 19 | 73 | 2.5 | 331 | 10146.1 | 7551.8 |
| Nov. | 13.5 | 68 | 1.9 | 32.6 | 10178.8 | 5759.2 |
| Dec. | 6.3 | 64 | 2.0 | 82.7 | 10208.6 | 3941.2 |

*Note: average values for air temperature (T), relative humidity (RH), wind speed (WS), pressure, and visibility and
accumulated monthly value for rainfall, respectively.



Table 2 Mean species levels for different seasons and different time of day and comparisons with other previous data
reported in typical regions in China.

| Species | Areas | Location | Period | The whole day | | | Day time (08:00-17:00) | | | Night time (18:00-07:00) | | |
|---|---|---|---|---|---|---|---|---|---|---|---|---|
| | | | | Mean | SD | Max | Mean | SD | Max | Mean | SD | Max |
| $PM_{2.5}$ (ug m$^{-3}$) | | This study | DJF | 74.2 | 49.3 | 406.4 | 75.1 | 50.5 | 406.4 | 73.6 | 48.4 | 325.5 |
| | | | MAM | 47.1 | 26.2 | 201.1 | 47.7 | 26.6 | 201.1 | 46.7 | 25.9 | 154.0 |
| | | | JJA | 34.6 | 22.5 | 181 | 35.1 | 25.7 | 181.0 | 34.3 | 20.0 | 139.6 |
| | | | SON | 52.5 | 34.4 | 272.4 | 51.7 | 33.3 | 238.1 | 53.1 | 35.1 | 272.4 |
| | YRD | [a] Xiacheng District, Hangzhou (Sep.-Nov. 2013) monthly mean: 69 µg m$^{-3}$ | | | | | | | | | | |
| | | [b] NRCS, Hangzhou (2012) annual mean: 50.0 µg m$^{-3}$ | | | | | | | | | | |
| | | [c] Hangzhou (Sep. 2010-Nov. 2011 during non-raining days) annual average:106-131 µg m$^{-3}$ | | | | | | | | | | |
| | | [d] Nine sites in Nanjing (2013) AM: 55-60 µg m$^{-3}$, JJA: 30-60 µg m$^{-3}$, SON: 55-85 µg m$^{-3}$ | | | | | | | | | | |
| | | [e] Nanjing (Mar. 2013-Feb. 2014) annual mean: 75± 50 µg m$^{-3}$ | | | | | | | | | | |
| | | [e] Shanghai (Mar. 2013-Feb. 2014) annual mean: 56 ± 41 µg m$^{-3}$ | | | | | | | | | | |
| | BTH | [e] Beijing (Mar. 2013-Feb. 2014) annual mean: 87±67 µg m$^{-3}$ | | | | | | | | | | |
| | PRD | [e] Guangzhou (Mar. 2013-Feb. 2014) annual mean: 52±28 µg m$^{-3}$ | | | | | | | | | | |
| $PM_{10}$ (ug m$^{-3}$) | | This study | DJF | 113.1 | 71.7 | 589.6 | 115.3 | 73.6 | 589.6 | 111.5 | 70.4 | 481.6 |
| | | | MAM | 77.1 | 42.3 | 484.1 | 79.3 | 41.0 | 249.1 | 75.6 | 43.2 | 484.1 |
| | | | JJA | 54.9 | 31.6 | 231.4 | 55.7 | 34.8 | 231.4 | 54.4 | 29.2 | 183.8 |
| | | | SON | 85.6 | 51.2 | 344.2 | 84.8 | 48.6 | 341.3 | 86.1 | 53.0 | 344.2 |
| | YRD | [e] Hangzhou (Mar. 2013-Feb. 2014) annual mean: 98 ± 59 µg m$^{-3}$ | | | | | | | | | | |
| | | [c] Hangzhou (Sep. 2010-Nov. 2011 during non-raining days) annual average: 127-158 µg m$^{-3}$ | | | | | | | | | | |
| | | [f] Hangzhou (Sep. 2001-Aug. 2002) annual mean: 119.2 µg m$^{-3}$ | | | | | | | | | | |
| | | [e] Nanjing (Mar. 2013-Feb. 2014) annual mean: 134 ± 73 µg m$^{-3}$ | | | | | | | | | | |
| | | [e] Shanghai (Mar. 2013-Feb. 2014) annual mean: 80 ± 47 µg m$^{-3}$ | | | | | | | | | | |
| | BTH | [e] Beijing (Mar. 2013-Feb. 2014) annual mean: 109±62 µg m$^{-3}$ | | | | | | | | | | |
| | PRD | [e] Guangzhou (Mar. 2013-Feb. 2014) annual mean: 72±35 µg m$^{-3}$ | | | | | | | | | | |
| $O_3$ (ppbv) | | This study | DJF | 13.8 | 13.1 | 70.9 | 17.7 | 14.1 | 70.9 | 10.2 | 10.9 | 58.5 |
| | | | MAM | 29.8 | 24.0 | 141.2 | 42.4 | 27.3 | 141.2 | 20.0 | 15.1 | 105.9 |
| | | | JJA | 31.3 | 26.0 | 145.4 | 48.8 | 26.6 | 145.4 | 18.2 | 15.8 | 118.7 |
| | YRD | | SON | 25.9 | 22.5 | 100.1 | 37.0 | 25.1 | 100.1 | 16.3 | 14.3 | 99.5 |
| | | [e] Hangzhou (Mar. 2013-Feb. 2014) annual mean: 44 ± 21 ppbv (8 h $O_3$) | | | | | | | | | | |
| | | [e] Nanjing (Mar. 2013-Feb. 2014) annual mean: 42 ± 20 ppbv (8 h $O_3$) | | | | | | | | | | |
| | | [e] Shanghai (Mar. 2013-Feb. 2014) annual mean: 48 ± 21 ppbv (8 h $O_3$) | | | | | | | | | | |
| | BTH | [e] Beijing (Mar. 2013-Feb. 2014) annual mean: 45 ± 27 ppbv (8 h $O_3$) | | | | | | | | | | |
| | PRD | [e] Guangzhou (Mar. 2013-Feb. 2014) annual mean: 45 ± 24 ppbv (8 h $O_3$) | | | | | | | | | | |
| $SO_2$ (ppbv) | YRD | This study | DJF | 14.5 | 10.2 | 71.2 | 16.2 | 10.2 | 71.2 | 13.3 | 10.2 | 64.6 |



| | | Season | | | | | | | | | |
|---|---|---|---|---|---|---|---|---|---|---|---|
| | | MAM | 11.3 | 9.1 | 75.1 | 11.7 | 9.6 | 75.1 | 11.0 | 8.7 | 59.3 |
| | | JJA | 8.6 | 6.5 | 51.0 | 8.0 | 6.3 | 51.0 | 9.0 | 6.6 | 46.7 |
| | | SON | 9.6 | 7.2 | 63.8 | 10.3 | 7.1 | 58.3 | 9.0 | 7.3 | 63.8 |

[a] Hangzhou Xiacheng District (12-19 Oct., 2013) daily mean: 5.7-9.7 ppbv

[e] Hangzhou (Mar. 2013-Feb. 2014) annual mean: 9 ±4 ppbv

[e] Nanjing (Mar. 2013-Feb. 2014) annual mean: 12 ± 6 ppbv

[e] Shanghai (Mar. 2013-Feb. 2014) annual mean: 7 ± 5 ppbv

BTH   [e] Beijing (Mar. 2013-Feb. 2014) annual mean: 9 ± 8 ppbv

PRD   [e] Guangzhou (Mar. 2013-Feb. 2014) annual mean: 7 ± 3 ppbv

**CO (ppmv)**

| | | | DJF | 1.4 | 0.7 | 3.8 | 1.4 | 0.7 | 3.3 | 1.4 | 0.7 | 3.8 |
|---|---|---|---|---|---|---|---|---|---|---|---|---|
| | | This study | MAM | 0.7 | 0.2 | 2.2 | 0.7 | 0.3 | 2.2 | 0.7 | 0.2 | 1.7 |
| | | | JJA | 0.5 | 0.2 | 2.0 | 0.5 | 0.2 | 1.9 | 0.5 | 0.2 | 2.0 |
| | YRD | | SON | 0.8 | 0.3 | 3.4 | 0.7 | 0.3 | 1.9 | 0.8 | 0.3 | 3.4 |

[e] Hangzhou (Mar. 2013-Feb. 2014) annual mean: 0.7 ±0.3 ppmv

[e] Nanjing (Mar. 2013-Feb. 2014) annual mean: 0.8 ±0.4 ppmv

[e] Shanghai (Mar. 2013-Feb. 2014) annual mean: 0.7 ±0.3 ppmv

BTH   [e] Beijing (Mar. 2013-Feb. 2014) annual mean: 1.1 ± 0.7 ppmv

PRD   [e] Guangzhou (Mar. 2013-Feb. 2014) annual mean: 0.8 ± 0.2 ppmv

**NO₂ (ppbv)**

| | | | DJF | 37.4 | 20.1 | 146.9 | 35.7 | 19.5 | 126.3 | 38.5 | 20.5 | 146.9 |
|---|---|---|---|---|---|---|---|---|---|---|---|---|
| | | This study | MAM | 28.7 | 12.9 | 94.8 | 25.3 | 12.1 | 94.8 | 31.0 | 12.9 | 87.4 |
| | | | JJA | 17.3 | 10.2 | 61.4 | 13.0 | 9.2 | 46.1 | 20.3 | 9.7 | 61.4 |
| | YRD | | SON | 28.4 | 15.2 | 94.1 | 25.1 | 13.3 | 86.2 | 30.7 | 16.0 | 94.1 |

[e] Hangzhou (Mar. 2013-Feb. 2014) annual mean: 13 ±9 ppbv

[e] Nanjing (Mar. 2013-Feb. 2014) annual mean: 26 ±11 ppbv

[e] Shanghai (Mar. 2013-Feb. 2014) annual mean: 20 ±9 ppbv

BTH   [e] Beijing (Mar. 2013-Feb. 2014) annual mean: 25 ±11 ppbv

PRD   [e] Guangzhou (Mar. 2013-Feb. 2014) annual mean: 24 ±10 ppbv

**NO_x (ppbv)**

| | | | DJF | 60.5 | 34.7 | 199.8 | 58.0 | 32.1 | 168.9 | 62.3 | 36.3 | 199.8 |
|---|---|---|---|---|---|---|---|---|---|---|---|---|
| | YRD | This study | MAM | 40.0 | 19.8 | 131.4 | 36.5 | 19.2 | 129.2 | 42.5 | 19.8 | 131.4 |
| | | | JJA | 24.3 | 14.8 | 99.6 | 18.6 | 14.1 | 99.6 | 28.2 | 14.0 | 83.1 |
| | | | SON | 41.0 | 24.3 | 153.4 | 36.6 | 21.1 | 123.7 | 44.2 | 25.8 | 153.4 |

**NO_y (ppbv)**

| | | | DJF | 84.7 | 48.4 | 295.2 | 82.4 | 44.6 | 263.7 | 86.4 | 51.1 | 295.2 |
|---|---|---|---|---|---|---|---|---|---|---|---|---|
| | | This study | MAM | 66.0 | 33.6 | 248.8 | 62.9 | 34.6 | 248.8 | 68.2 | 32.8 | 204.1 |
| | YRD | | JJA | 43.6 | 27.6 | 259.5 | 36.8 | 29.3 | 259.5 | 48.5 | 25.2 | 167.7 |
| | | | SON | 70.2 | 37.9 | 319.3 | 65.5 | 35.6 | 319.3 | 73.6 | 39.1 | 251.8 |

[g] Nanjing SORPES 2013 monthly mean: 30-70 ppbv

[h] Shanghai May-June 2005 daily mean: 24-39 ppbv





| | |
|---|---|
| BTH | [a] Beijing 2011-2015 annual mean: 54.6 ± 4.7 ppbv |
| YRD | [h] Guangzhou Apr.-May 2004: 24-52 ppbv |

[a] Wu et al. (2016a); [b] Qi et al. (2015); [c] Sun et al. (2013); [d] Chen et al. (2016); [e] Wang et al. (2014); [f] Cao et al. (2009); [g] Ding
et al. (2013); [h] Xue et al. (2014)



Table 3 Mean concentrations of $PM_{2.5}$ ($\mu g\ m^{-3}$) and other trace gases (ppmv unit for CO but ppbv for other gases) in the
identified trajectory clusters within four season period, together with the percentages of each trajectory cluster.

| Season | Cluster | Percent (%) | $PM_{2.5}$ | $O_3$ | $SO_2$ | CO | $NO_x$ |
|---|---|---|---|---|---|---|---|
| Spring | 1 | 12.05 | 45.0 | 28.3 | 10.7 | 0.7 | 38.3 |
| | 2 | 16.58 | 44.3 | 31.6 | 13.2 | 0.7 | 39.1 |
| | 3 | 16.03 | 35.3 | 30.5 | 9.7 | 0.6 | 34.5 |
| | 4 | 42.66 | 52.4 | 23.2 | 11.4 | 0.8 | 42.5 |
| | 5 | 5.53 | 38.2 | 34.2 | 11.2 | 0.7 | 37.9 |
| | 6 | 7.16 | 58.1 | 34.2 | 11.9 | 0.8 | 43.8 |
| Summer | 1 | 8.42 | 51.5 | 24.6 | 7.9 | 0.8 | 29.2 |
| | 2 | 8.61 | 34.2 | 35.2 | 9.2 | 0.5 | 22.8 |
| | 3 | 22.55 | 24.0 | 28.7 | 7.9 | 0.4 | 21.7 |
| | 4 | 31.34 | 38.2 | 36.8 | 9.1 | 0.5 | 24.4 |
| | 5 | 19.38 | 38.7 | 27.2 | 8.9 | 0.6 | 28.7 |
| | 6 | 9.69 | 22.4 | 26.7 | 7.5 | 0.4 | 17.6 |
| Autumn | 1 | 23.63 | 42.1 | 27.4 | 9.9 | 0.7 | 36.9 |
| | 2 | 32.51 | 50.7 | 24.6 | 8.2 | 0.8 | 39.4 |
| | 3 | 8.33 | 21.7 | 19.8 | 8.0 | 0.5 | 22.0 |
| | 4 | 7.78 | 68.6 | 34.8 | 8.4 | 0.8 | 38.8 |
| | 5 | 11.90 | 49.9 | 22.6 | 10.1 | 0.7 | 40.8 |
| | 6 | 15.84 | 79.6 | 21.6 | 12.9 | 0.9 | 62.0 |
| Winter | 1 | 7.13 | 60.9 | 16.6 | 15.4 | 1.3 | 53.7 |
| | 2 | 24.26 | 83.3 | 14.4 | 15.9 | 1.4 | 65.4 |
| | 3 | 16.39 | 47.3 | 14.0 | 11.9 | 1.1 | 42.7 |
| | 4 | 21.76 | 75.9 | 11.9 | 13.5 | 1.5 | 63.1 |
| | 5 | 16.76 | 67.0 | 11.7 | 13.1 | 1.5 | 53.7 |
| | 6 | 13.70 | 102.1 | 14.4 | 16.9 | 1.4 | 81.0 |





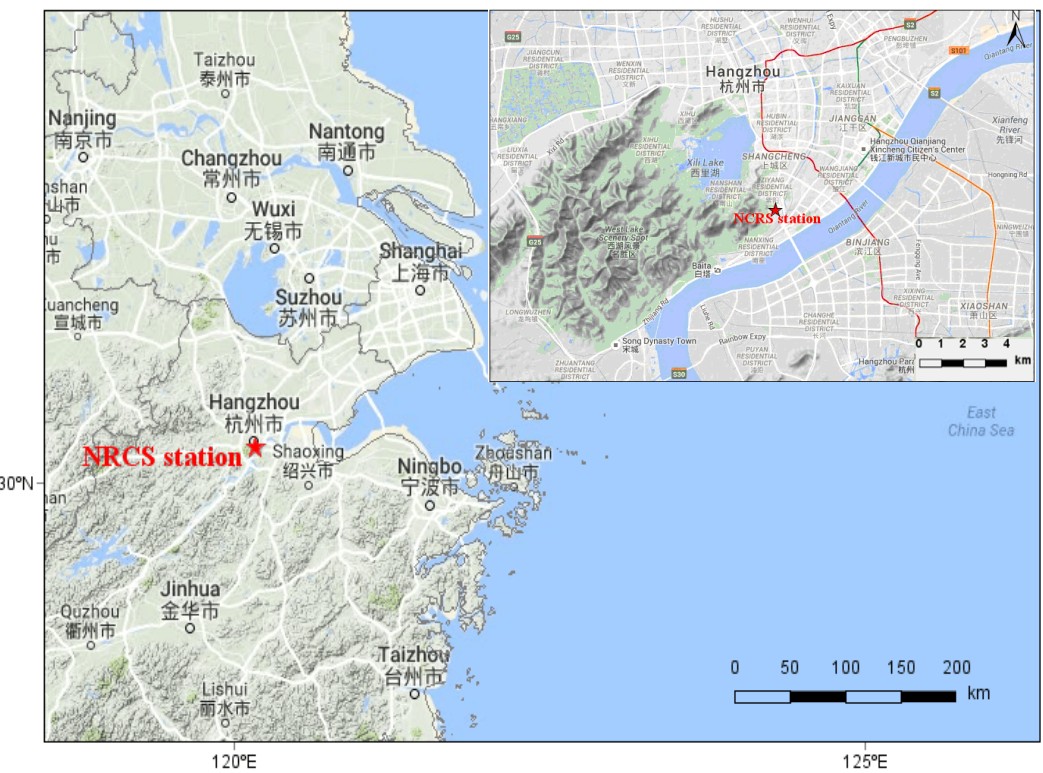


Fig. 1. Location of NRCS station in YRD region (left) and in the city of Hangzhou (right top).




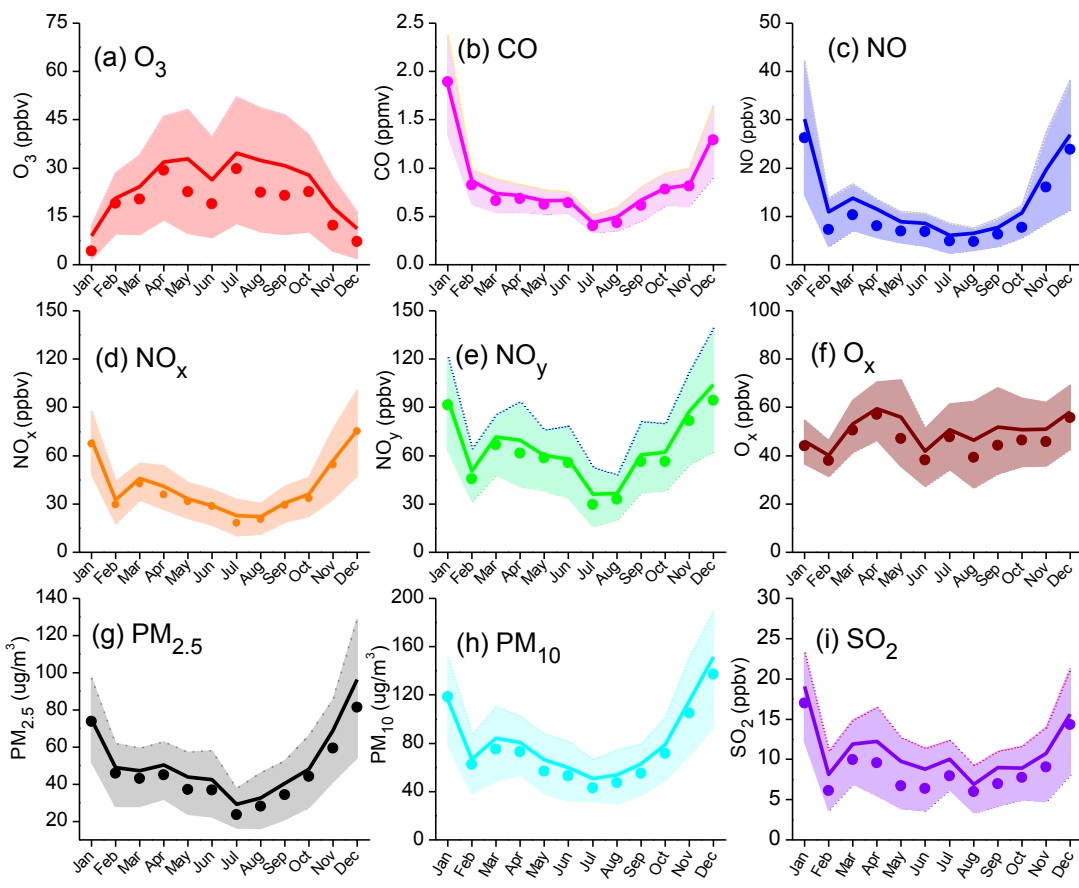


Fig. 2. Seasonal variations of atmospheric $O_3$ (a), CO (b), NO (c), $NO_x$ (d), $NO_y$ (e), $O_x$ (f), $PM_{2.5}$ (g), $PM_{10}$ (h), and $SO_2$ (i).
Bold solid lines are the monthly averages, solid circles are the median values, and thin lines represent percentiles of 75% and

25%.



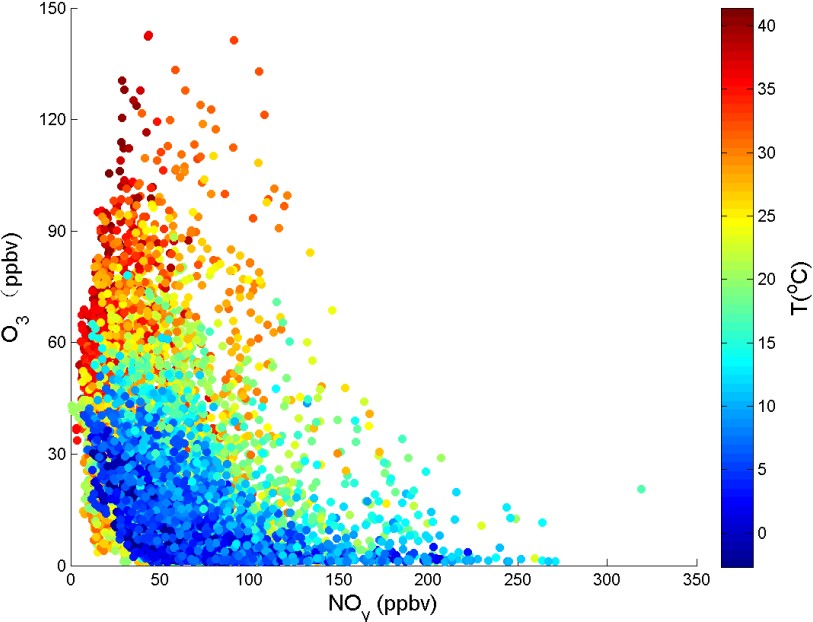


Fig. 3. Scatter plots of NO$_y$ with O$_3$ coded with air temperature

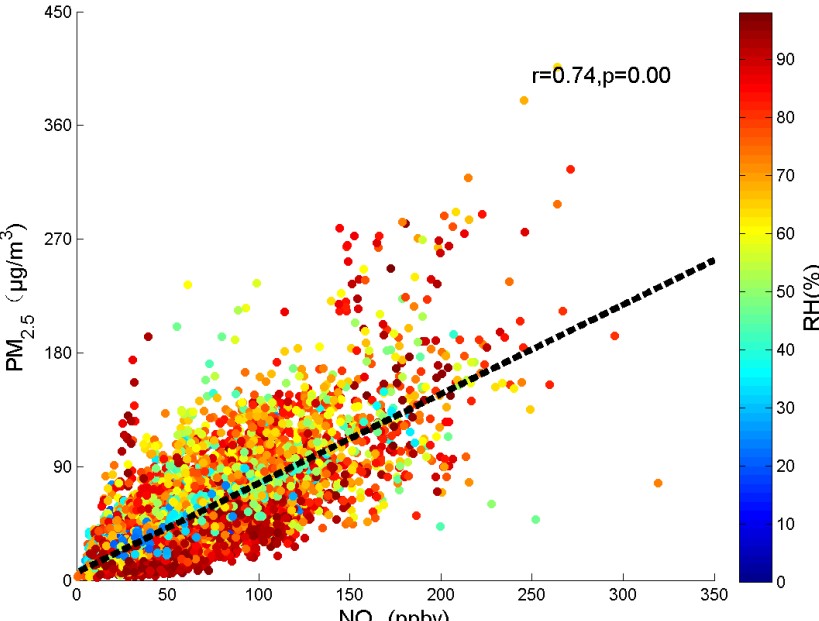


Fig. 4. Scatter plots of NO$_y$ with PM$_{2.5}$ coded with relative humidity




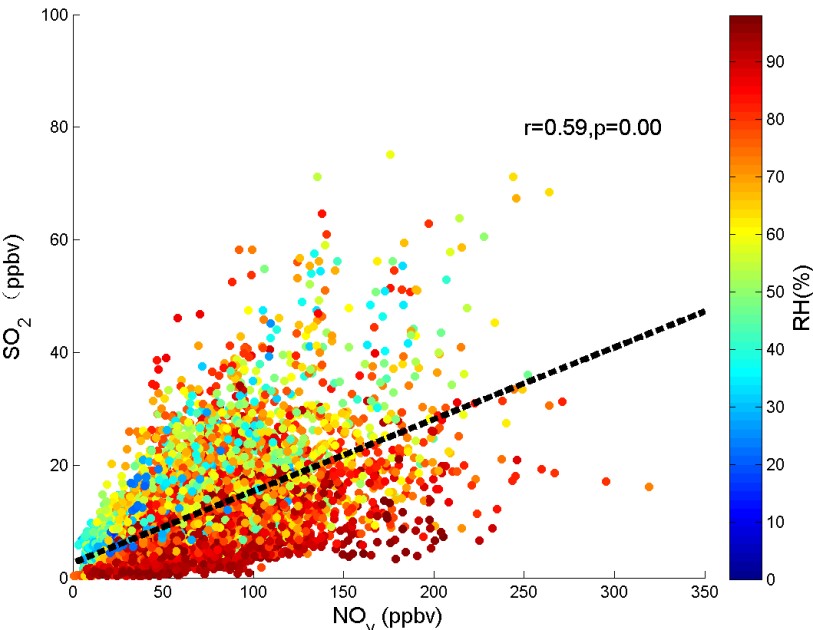

Fig. 5. Scatter plots of $NO_y$ with $SO_2$ coded with relative humidity

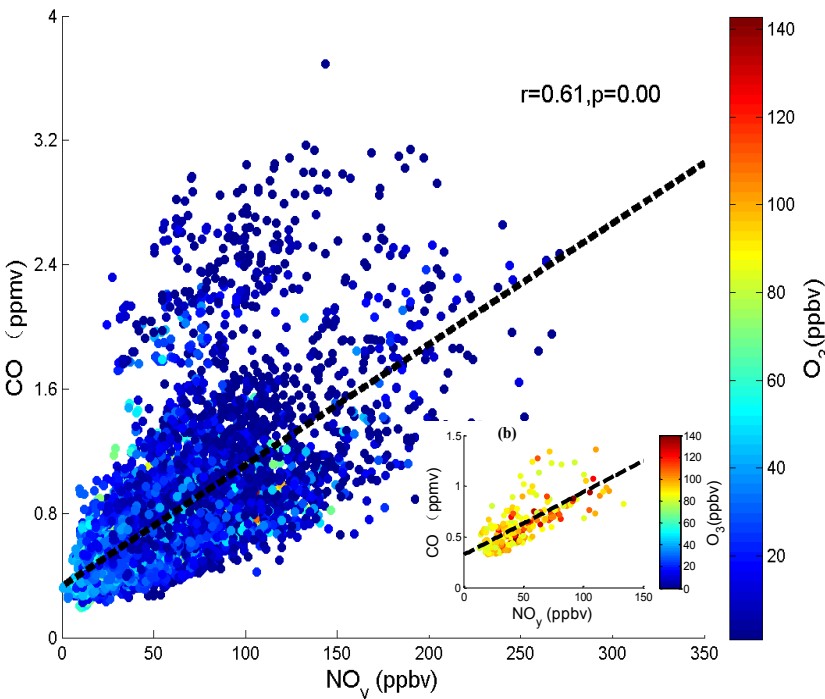

Fig. 6. Scatter plots of $NO_y$ with CO coded with $O_3$ mixing ratios, along the subpicture (b) showing the scatter with $O_3$
mixing ratios above 80 ppbv.




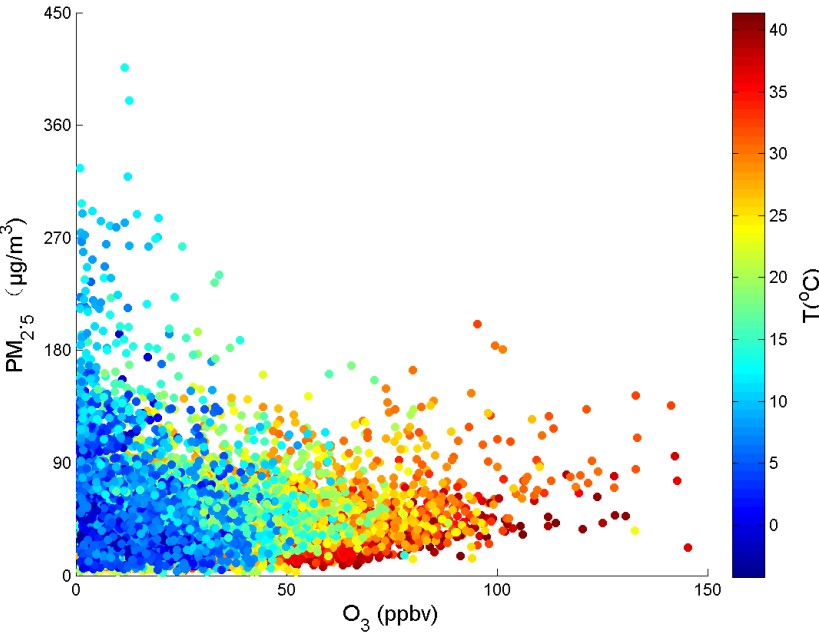


Fig. 7. Scatter plots of $O_3$ with $PM_{2.5}$ coded with air temperature






Fig. 8. Seasonal cluster analysis of the 72-h air mass back trajectories starting at 100 m from NRCS site in Hangzhou.





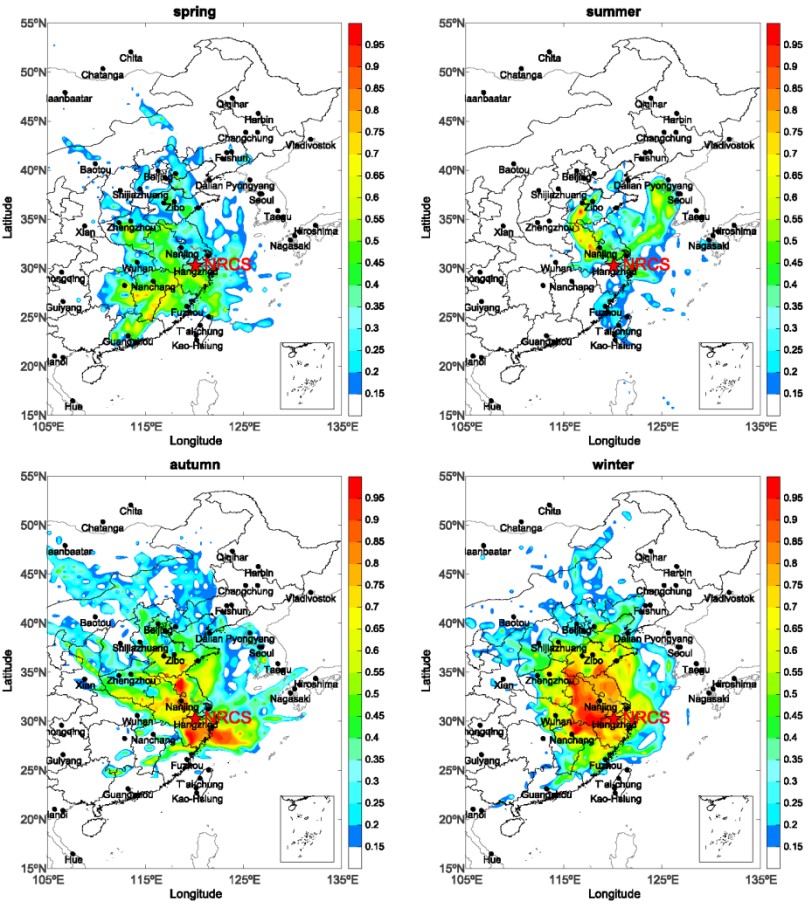


Fig. 9a. Seasonal weighted potential source contribution function (WPSCF) maps of PM$_{2.5}$ in Hangzhou. The sampling site is

marked in pentacle and the WPSCF values are displayed in color.

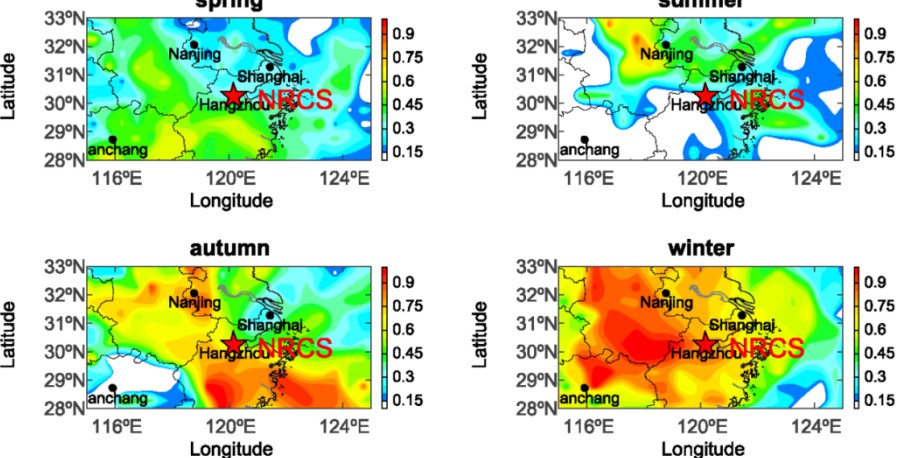


Fig. 9b. The zoomed view of Fig. 9a.






Fig. 9c. Seasonal and spatial distributions of PM$_{2.5}$ emissions (kg km$^2$ mon$^{-1}$) at the surface layer in China. The sampling site is marked in pentacle.





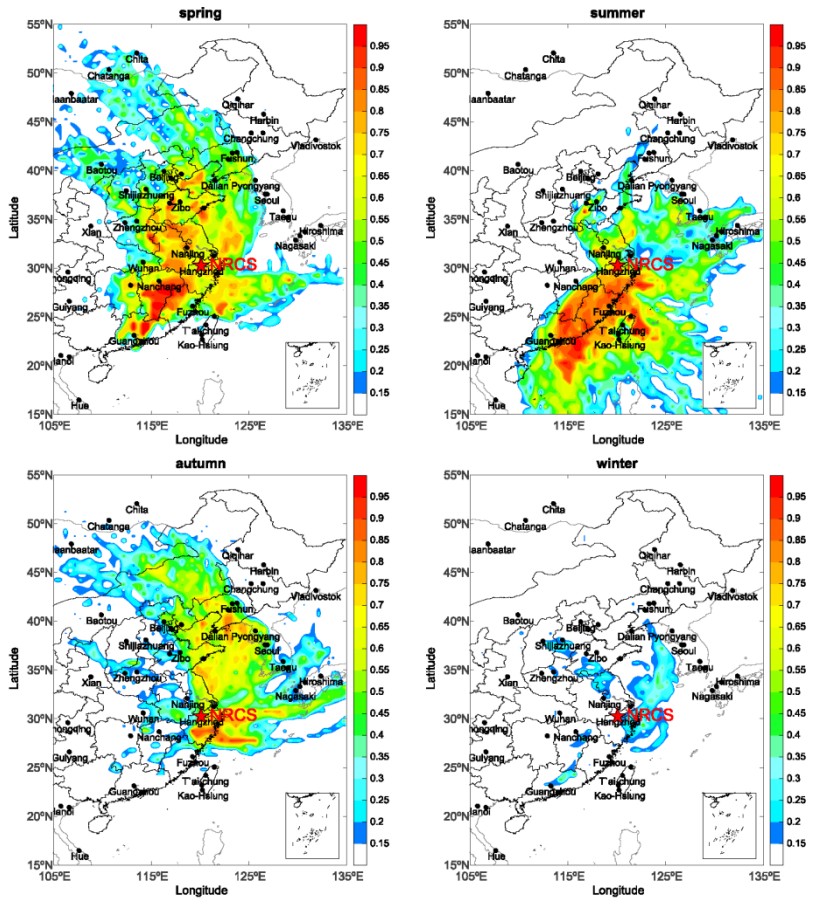


Fig. 10a. Same as Fig. 9a but for $O_3$

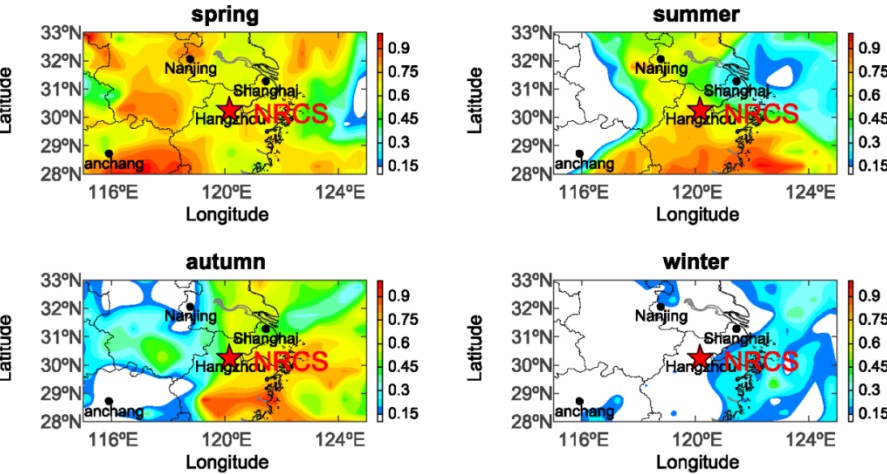


Fig. 10b The zoomed view of Fig. 10a








Fig. 11. Time series of meteorological parameters and chemical species before, during, and after haze period. The gray

shaded area indicates the Phase I (28 Nov.-3 Dec.) and Ⅱ (10-12 Dec.) and the orange shaded area represents haze events

Phase Ⅲ (2-9 Dec.) and Ⅳ (13-15 Dec.).





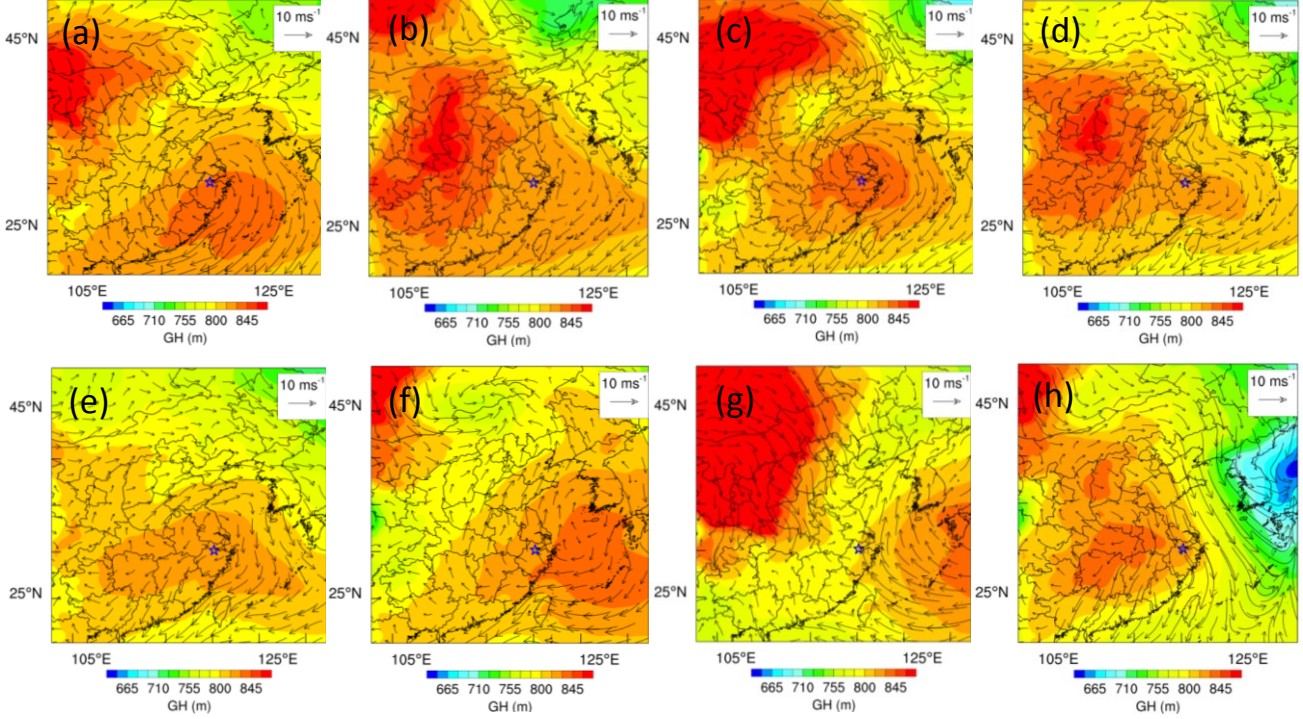

Fig. 12. The Geopotential Height Field (GH) (indicated by color bars) and Wind Field (WF) (black vectors) for 925 hPa at 20:00 LT during 2-9 December, 2013. Fig.12a-d and Fig. 12e-h represent for 2-5 December and 6-9 December from left to right on the top and bottom, respectively. The NRCS station was marked by pentagram.







779

Fig. 13. Same as Fig. 11 but during photochemical pollution period. The orange shaded area represents the Phase I (28-30

May and 20-22 June), the cyan shaded area indicates the Phase Ⅱ (9-12 July), and the other area represents the Phase Ⅲ (1-3

May, 20-22 May, and 9-11 August)