# Peer review of "Characterization of atmospheric trace gases and particulate matters in Hangzhou, China"

_Atmospheric Chemistry and Physics, 2017_

## Referee Comment (RC1) · Anonymous Referee #1 · 26 Oct 2017

This manuscript reports one-year continuous measurements of trace gases and particulate matters at a National Reference Climatological Station in Hangzhou, southern Yangtze River Delta region. The data were analyzed in terms of seasonal variations, interspecies correlations, and potential contributions from local emissions and regional transport. The measurement data of the present study are much valuable, and the analysis and interpretation of the data are fairly well. Thus, it is recommended that this manuscript can be considered for publication after the following comments being addressed.

Specific Comments:

1. Overall, the interpretation and analyses of the measurement results are fairly well, but there is lack of comparison with the other studies and importance or implications

of the present study. To date there have been many measurement studies in the YRD region, such as at Lin'an, Shanghai and Nanjing. The authors should point to the new findings or difference between this new piece of work and the previous studies.

2. The first paragraph of the Introduction section contains a lot of very basic information on the individual trace gases. I presume that the readership of the Journal should be expertise of this field, and suggest the authors to remove (or shorten) such general description and just focus on the key knowledge gaps and motivation of the study in the Introduction part.

3. Page 2, Line 52: intermediates/products

4. Page 2, Line 55: and/or

5. Page 3, Lines 80-81: it is not clear what the "large knowledge gap and discrepancy" means. Please elaborate more about the knowledge gap.

6. Line 3, Line 89: Experiment and meteorological conditions

7. Line 3, Line 93: please provide the standard deviations for the average temperature, RH and rainfall.

8. Section 2.1: the authors need clearly state the type (e.g., urban, suburban or rural) of the study site. What are the major emission sources surrounding the site?

9. Section 2.2, on the measurements of NO2 and CO: what kind of converter was used for the conversion from NO2 to NO? Is there auto-zero or auto-reference function for the CO analyzer, and what is the time frequency for the auto-zeroing during the campaign?

10. Page 5, Line 132: change "showed" to "found"

11. Page 6, Lines 165-166: it would be better to use daily maximum 8-hour average ozone concentration for estimating the ozone exceedance days.

[Figure]

12. Page 6, Line 172: change "between" to "among"

13. Page 6, Lines 180-182: higher NOy levels cannot indicate more photochemical oxidation of NOx, as NOy is the total nitrogen oxides including both NOx and its oxidation products (NOz). More NOy suggests the more abundance of nitrogen oxides in Hangzhou.

14. Page 7, Lines 188-191: some studies have also investigated the seasonal variations of O3 in Hong Kong and North China, and the authors should acknowledge these earlier studies.

Xue, L. K., Wang T., Louie P. K. K., Luk, C. W. Y., Blake D. R., Xu Z.: Increasing external effects negate local efforts to control ozone air pollution: a case study of Hong Kong and implications for other Chinese cities, Environ. Sci. Tech., 48(18), 10769-10775, 2014.

Sun L., Xue L. K., Wang T., Gao J., Ding A. J., Cooper O. R., Lin M. Y., Xu P. J., Wang Z., Wang X. F., Wen L., Zhu Y. H., Chen T. S., Yang L. X., Wang Y., Chen J. M., Wang W. X.: Significant increase of summertime ozone at Mount Tai in Central Eastern China, Atmos. Chem. Phys., 16, 10637-10650, 2016.

15. Page 7, Line 192: Xianlin?

16. Page 8, Lines 227-228: revise this sentence

17. Page 9, Line 257: color-coded

18. Page 9, Line 258: led to

19. Page 9, Line 275: change "in addition to" to "in view of"

20. Page 10, Line 297: pay attention to

21. Page 11, Lines 329-330: why the air masses coming from open seas contained higher concentrations of NOx and O3? The authors need elaborate more about the

reason for this interesting result.

22. Page 12, Line 359: long distance transport

23. Figures 3-7: it would be better to combine these figures into one figure.

---

## Referee Comment (RC2) · Anonymous Referee #2 · 30 Oct 2017

The manuscript 'Characterization of atmospheric trace gases and particle matters in Hangzhou, China' by G. Zhang et al. reports the observational results from one-year monitoring of several trace gases and particulate matter at an urban site in the YRD region. The characteristics of these trace gases and particulate matter are discussed in association with meteorological conditions. Process analysis is also performed for case studies under photochemical pollution and haze condition. The measurement data are valuable, but the manuscript needs to be more concise and more logically structured. Further proofreading is also needed to correct grammar mistakes and inappropriate description.

Specific comments:

1. 'Particle matter' is used almost through the entire manuscript, it should be particulate

matter.

2. Was the air sample dried when measuring PM2.5? How about the drying system?

3. What is the temporal resolution of the meteorological data in the HYSPLIT model? Will the temporal resolution and also the spatial resolution as 0.5o×0.5 o influence your conclusions?

4. P9 L246-249, the author suggested comparable photochemical levels in different regions only based on measurements of NO2 and O3, I am afraid it is insufficient to draw this conclusion.

5. The discussion on NOx or VOCs limitation of ozone photochemical production is based on measured CO. The author stated that VOCs and CO share common origins and play similar roles in ozone production in this region. Is there any data or previous study in this region to support this assumption?

6. The correlations of O3 and PM2.5 in warm and cold seasons were analyzed. The author attributed the positive correlation in warm seasons to secondary aerosol formation under high O3 levels and negative correlation in cold seasons to NO titration effect. However, the ambient level of either O3 or PM2.5 is a result of emission, sinks, physical processes and complicated chemical reactions. The explanation has no solid foundation and also needs other supporting data.

7. The backward trajectory and PSCF analysis is not suitable for short-lived species such as O3 and is specially not suitable in urban area with high local emission. So it's strange that those clean mountain area in south of Hangzhou could have more contributions? As well as that air masses coming from open seas contained higher concentrations of NOx and O3?

---

## Author Comment (AC1) · 30 Nov 2017

**Response to Anonymous Referee #1**

This manuscript reports one-year continuous measurements of trace gases and particulate matters at a National Reference Climatological Station in Hangzhou, southern Yangtze River Delta region. The data were analyzed in terms of seasonal variations, interspecies correlations, and potential contributions from local emissions and regional transport. The measurement data of the present study are much valuable, and the analysis and interpretation of the data are fairly well. Thus, it is recommended that this manuscript can be considered for publication after the following comments being addressed.

**Response:** Thanks for your recognition and positive comments on our manuscript. According to your suggestions, we made the corresponding corrections in the revised manuscript. We expect this version would meet the requirement for publication in ACP.

Specific Comments:

1. Overall, the interpretation and analyses of the measurement results are fairly well, but there is lack of comparison with the other studies and importance or implications of the present study. To date there have been many measurement studies in the YRD region, such as at Lin'an, Shanghai and Nanjing. The authors should point to the new findings or difference between this new piece of work and the previous studies.

**Response:** Thank you very much for the suggestions about making comparison with the other studies. As depicted in the revised manuscript, we have elaborated more examples about the knowledge gap between this work and the previous studies and further pointed out our new finding.

2. The first paragraph of the Introduction section contains a lot of very basic information on the individual trace gases. I presume that the readership of the Journal should be expertise of this field, and suggest the authors to remove (or shorten) such general description and just focus on the key knowledge gaps and motivation of the study in the Introduction part.

**Response:** Yes, you are right. According to your comments, we removed some basic introduction related to the well-known trace gases including $NO_x$, CO, and $SO_2$ in the first paragraph in the revised manuscript.

3. Page 2, Line 52: intermediates/products

**Response:** As you suggested, we made corrections as "intermediates/products" in the revised manuscript.

4. Page 2, Line 55: and/or

**Response:** As you suggested, we replaced it with "and/or" in the revised manuscript.

5. Page 3, Lines 80-81: it is not clear what the "large knowledge gap and discrepancy" means. Please

elaborate more about the knowledge gap.

**Response:** Thanks for your valuable comments about "large knowledge gap and discrepancy". As mentioned in Response 2 above, we elaborated more examples about the knowledge gap in the revised manuscript.

6. Line 3, Line 89: Experiment and meteorological conditions

**Response:** In the revised manuscript, we changed it with "Experiment and meteorological conditions" as you suggested.

7. Line 3, Line 93: please provide the standard deviations for the average temperature, RH and rainfall.

**Response:** Thanks for your mention on this expression to improve our manuscript. As you suggested, we added the standard deviations for the average air temperature and RH. However, as shown in Table 1, we used the accumulated monthly value for rainfall (not the average) and thus it has no standard deviations.

8. Section 2.1: the authors need clearly state the type (e.g., urban, suburban or rural) of the study site. What are the major emission sources surrounding the site?

**Response:** We have stated the type of our study site (urban site) in the revised manuscript. With respect to the major emission sources surrounding this site, we have made the specific introduction as below in the revised manuscript.

"As a typical urban site, NRCS station is situated in the commercial and residential areas in the southern Hangzhou and thus it's characterized as a polluted receptor site as it receives local urban plumes and regional air masses from the YRD region when northwesterly wind prevails. In addition, as the right top map shown in Fig. 1, the site is adjacent to Prince Bay Park (area, 0.8 km$^2$) and situated in the northeastern part of West Lake famous scenic spot (area, 49 km$^2$). Therefore it can also capture the signature of vegetation emission in urban Hangzhou under southwesterly winds. Moreover, there are no local industrial pollution sources around the site. In brief, this site can be representative of urban Hangzhou."

9. Section 2.2, on the measurements of $NO_2$ and CO: what kind of converter was used for the conversion from $NO_2$ to NO? Is there auto-zero or auto-reference function for the CO analyzer, and what is the time frequency for the auto-zeroing during the campaign?

**Response:** Your suggestions are very important and valuable. In our study, we used internal and external MoO converter for the conversion from $NO_2$ and NOz to NO, respectively. For CO analyzer,

there was auto-zero function and its time frequency was every 6 h during the campaign. The corresponding corrections were made in the revised manuscript, as shown below.

"NO and $NO_x$ were detected by a chemiluminescence analyzer coupled with an internal MoO catalytic converter (TEI, 42i). Note that the differentiated value of $NO_2$ from $NO_x$ and NO represents the upper limit concentration of atmospheric $NO_2$ due to the interference of other nitrogen-containing components (e.g., PAN, $HNO_3$, and HONO) in the conversion. Similar with $NO_x$, $NO_y$ was also detected by a chemiluminescence analyzer (TEI 42i-Y) but equipped with an external MoO catalytic converter. CO was monitored with a gas filter correlation, infrared absorption analyser (TEI, 48i), with automatic zeroing every 6 hours."

14. Page 7, Lines 188-191: some studies have also investigated the seasonal variations of $O_3$ in Hong Kong and North China, and the authors should acknowledge these earlier studies.
**Response:** As you suggested, we have added the two earlier studies in the revised manuscript.

15. Page 7, Line 192: Xianlin?
**Response:** Yes, you are right. We are sorry for this mistake. This site is "Xianlin"

16. Page 8, Lines 227-228: revise this sentence
**Response:** According to your comment, we have changed this sentence with "In summer (Fig. 2f), an abnormally high level of $O_x$ was found in winter with low $O_3$." in the revised manuscript.

17. Page 9, Line 257: color-coded
**Response:** According to your comment, we have replaced "coded" with "color-coded" in the revised manuscript.

18. Page 9, Line 258: led to
**Response:** According to your comment, we have replaced it with "led to" in the revised manuscript.

19. Page 9, Line 275: change "in addition to" to "in view of"
**Response:** According to your comment, we have changed "in addition to" with "in view of" in the revised manuscript.

20. Page 10, Line 297: pay attention to

**Response:** According to your comment, we have replaced it with "pay attention to" in the revised manuscript.

21. Page 11, Lines 329-330: why the air masses coming from open seas contained higher concentrations of $NO_x$ and $O_3$? The authors need elaborate more about the reason for this interesting result.

**Response:** At first, we are so sorry for the incorrectly expression "long transports from Yellow Sea, East Sea, and South Sea were also important potential sources for $NO_x$ and $O_3$" in the initial manuscript. After careful examination, we found that air masses with the seemed high WPSCF values for $NO_x$ were not originating far from these open seas. They were just the broadening "tails" of high values contained in the areas with intensive anthropogenic $NO_x$ emissions from inland well-industrialized cities. This phenomenon was also found in other studies by using trajectory statistical method (Riuttanen et al., 2013).

Similar with $NO_x$, air masses containing the high WPSCF values of $O_3$ also didn't come from the open seas. Indeed, as depicted in the manuscript, such air masses were mostly from the offshore area of East China Sea, Yellow China Sea, or South China, respectively on southeastern Zhejiang, Jiangsu, and Fujian Province. We speculated the recirculation of pollutants by sea- and land-breeze circulations around the cities along the YRD and Hangzhou Bay which has been confirmed by Li et al. (2015, 2016b), was largely responsible for the increased concentration of $O_3$ at NRCS site.

Also, such an increase in $O_3$ concentrations in urbanized coastal areas have been observed and modeled in a number of studies (Oh et al., 2006; Levy et al., 2008; Martins et al., 2012). Moreover, to further judge whether air masses came from open seas contained higher concentrations of $NO_x$ and $O_3$, we used the results of MOZART-4/GEOS-5 simulation to draw the distribution maps of $NO_x$ and $O_3$ concentrations within the identical domain (15-55 $^o$N and 105-135 $^o$E) with WPSCF analysis. As clearly seen from the Figure 1 below, high $NO_x$ mainly distributed in terrestrial regions, especially in industrialized cities, but very low $NO_x$ were found in open seas. In comparison, significantly high $O_3$ were elucidated covering the offshore regions of either East China Sea, Yellow China Sea, or South China Sea (Fig. 2). Then, along with the seasonal cluster analysis of back trajectories from NRCS site in Hangzhou, it's well confirmed that our speculation about the contribution of the recirculation of pollutants by sea- and land-breeze circulations in the offshore area to the observed $O_3$ at NRCS site.

[Figure]

Figure 1 Seasonal and spatial distributions of NO$_x$ volume mixing ratio (VMR) simulated by MOZART-4/GEOS-5. The sample site is marked in pentacle.

[Figure]

Figure 2 Seasonal and spatial distributions of O$_3$ volume mixing ratio (VMR) simulated by MOZART-4/GEOS-5. The sample site is marked in pentacle.

**Reference**

Levy, I., Dayan, U., and Mahrer, Y.: A five-year study of coastal recirculation and its effect on air pollutants over the east Mediterranean region, J. Geophys. Res., 113, D16121, 2008.

Li, M. M., Mao, Z. C., Song, Y., Liu, M. X., and Huang, X.: Impact of the decadal urbanization on thermally induced circulations in eastern China, J. Appl. Meteorol. Clim., 54, 259-282, 2015.

Li, M. M., Song, Y., Mao, Z. C., Liu, M. X., and Huang, X.: Impact of thermal circulations induced by urbanization on ozone formation in the Pearl River Delta, China, Atmos. Environ., 127, 382-392, 2016b.

Martins, D. K., Stauffer, R. M., Thompson, A. M., Knepp, T. N., and Pippin, M.: Surface ozone at a coastal suburban site in 2009 and 2010: relationships to chemical and meteorological processes, J. Geophys. Res., 117, D5, 5306, 2012.

Oh, I. B., Kim, Y. K., Lee, H. W., and Kim, C. H.: An observational and numerical study of the effects

of the late sea breeze on ozone distributions in the Busan metropolitan area, Korea, Atmos. Environ., 40, 1284-1298, 2006.

Riuttanen, L., Hulkkonen, M., Dal Maso, M., Junninen, H., and Kulmala, M.: Trajectory analysis of atmospheric transport of fine particles, $SO_2$, $NO_x$ and $O_3$ to the SMEAR II station in Finland in 1996-2008, Atmos. Chem. Phys., 13, 2153-2164, 2013.

22. Page 12, Line 359: long distance transport

**Response:** According to your comment, we have replaced it with "long distance transport" in the revised manuscript.

23. Figures 3-7: it would be better to combine these figures into one figure.

**Response:** According to your comment, we have combined the four figures, the scatter plots of $NO_y$-$O_3$ coded with air temperature (a), $NO_y$-$PM_{2.5}$ coded with relative humidity (b), $NO_y$-$SO_2$ coded with relative humidity (c), and $O_3$-$PM_{2.5}$ coded with air temperature (d), into one figure as Fig. 3 shown in the revised manuscript. In order to facilitate the clear view of the subpicture showing the scatter with $O_3$ mixing ratios above 80 ppbv, we keep the scatter plots of $NO_y$ with CO in a single figure as Fig. 4 shown in the revised manuscript.

[Figure]

Fig. 4. Scatter plots of $NO_y$ with CO coded with $O_3$ mixing ratios, along the subpicture (b) showing the scatter with $O_3$ mixing ratios above 80 ppbv.

---

## Author Comment (AC2) · 30 Nov 2017

**Response to Anonymous Referee #2**

The manuscript 'Characterization of atmospheric trace gases and particle matters in Hangzhou, China' by G. Zhang et al. reports the observational results from one-year monitoring of several trace gases and particulate matter at an urban site in the YRD region. The characteristics of these trace gases and particulate matter are discussed in association with meteorological conditions. Process analysis is also performed for case studies under photochemical pollution and haze condition. The measurement data are valuable, but the manuscript needs to be more concise and more logically structured. Further proofreading is also needed to correct grammar mistakes and inappropriate description.

**Response:** Thanks for your approval and presenting the valuable comments on our manuscript. According to your suggestions, we restructured our manuscript logically, shortened some redundant description and corrected grammar mistakes in the revised version. We expect this version would meet the requirement for publication in ACP.

Specific comments:

1. 'Particle matter' is used almost through the entire manuscript, it should be particulate matter.

**Response:** According to your suggestion, we replaced "particle matter" with "particulate matter" in the revised manuscript.

2. Was the air sample dried when measuring $PM_{2.5}$? How about the drying system?

**Response:** As described in the manuscript, ambient $PM_{2.5}$ samples were collected using co-located Thermo Scientific (formerly R&P) Model 1405D samplers. This sampler has no dried unit in our study.

3. What is the temporal resolution of the meteorological data in the HYSPLIT model? Will the temporal resolution and also the spatial resolution as $0.5^{o} \times 0.5^{o}$ influence your conclusions?

**Response:** As described in Section 2.4.1 in the manuscript, the six hourly final archive data with $1^{o} \times 1^{o}$ spatial resolution were obtained from the National Center for Environmental Prediction's Global Data Assimilation System (GDAS) wind field reanalysis. Such designated data have been widely used in numerous previous studies (Li et al., 2015; Yu et al., 2014; Zhang et al., 2013). As you know, the numbers of back trajectories starting from a selected site during the appointed period are probably dependent of temporal and spatial resolution of the meteorological data. Nevertheless, the trajectory cluster analysis is based on the statistical results of air masses back trajectories, and it should don't change a lot. Thus they don't lead to a significant effect on the conclusion.

**References:**

Li, P. F., Yan, R. C., Yu, S. C., Wang, S., Liu, W. P., and Bao, H. M.: Reinstate regional transport of $PM_{2.5}$ as a major cause of severe haze in Beijing, Proc. Natl. Acad. Sci., 112(21), 2739-2740, 2015.

Yu, S. C., Zhang, Q. Y., Yan, R. C., Wang, S., Li, P. F., Chen, B. X., Liu, W. P., and Zhang, X. Y.: Origin of air pollution during a weekly heavy haze episode in Hangzhou, China, Environ. Chem. Lett., 12, 543-550, 2014.

Zhang, R, Jing, J., Tao, J., Hsu, S. C., Wang, G., Cao, J., Lee, C. S. L., Zhu, L., Chen, Z., Zhao, Y., and Shen, Z.: Chemical characterization and source apportionment of PM2.5 in Beijing: seasonal perspective, Atmos. Chem. Phys., 13, 7053-7074, 2013.

4. P9 L246-249, the author suggested comparable photochemical levels in different regions only based on measurements of $NO_2$ and $O_3$, I am afraid it is insufficient to draw this conclusion.

**Response:** After careful examination throughout the manuscript, we didn't find these sentences in this version of the manuscript. These sentences were possibly included in the previous version and have been removed in this version.

5. The discussion on $NO_x$ or VOCs limitation of ozone photochemical production is based on measured CO. The author stated that VOCs and CO share common origins and play similar roles in ozone production in this region. Is there any data or previous study in this region to support this assumption?

**Response:** For VOCs and CO in the typical urban regions, their common origin and similar behavior in ozone production have been explicitly elucidated (Atkinson, 2000) and widely validated in the previous studies (Baker et al., 2008; Schneidemesser et al., 2010; Ding et al., 2013). Moreover, based on the data of VOCs and CO obtained at Lin'an site, a regional station located in the east Zhejiang Province (50 km away from Hangzhou) in eastern China, Guo et al. (2004) found the common sources of VOCs and CO were vehicle emissions and biofuel burning, biomass burning and industrial emissions. In addition, we Therefore, we added the previous publications behind this sentence to support our assumption in the revised manuscript.

**References:**

Atkinson, R.: Atmospheric chemistry of VOCs and $NO_x$, Atmos. Environ., 34, 2063–2101, 2000.

Baker, A. K., Beyersdorf, A. J., Doezema, L. A., Katzenstein, A. K., Meinardi, S., Simpson, I. J., Blake, D. R., and Rowland, F. S.: Measurements of nonmethane hydrocarbons in 28 United States cities, Atmos. Environ., 42, 170-182, 2008.

Ding, A. J., Fu, C. B., Yang, X. Q., Sun, J. N., Zheng, L. F., Xie, Y. N., Herrmann, E., Nie, W., Petäjä, T., Kerminen, V. M., and Kulmala, M.: Ozone and fine particulate in the western Yangtze River Delta: an overview of 1 yr data at the SORPES station, Atmos. Chem. Phys., 13, 5813-5830, 2013.

Guo, H., Wang, T., Simpson, I. J., Blake, D. R., Yu, X. M., Kwok, Y. H., and Li, Y. S.: Source contributions to ambient VOCs and CO at a rural site in eastern China, 38(27), 4551-4560, 2004.

6. The correlations of $O_3$ and $PM_{2.5}$ in warm and cold seasons were analyzed. The author attributed the positive correlation in warm seasons to secondary aerosol formation under high $O_3$ levels and negative

correlation in cold seasons to NO titration effect. However, the ambient level of either $O_3$ or $PM_{2.5}$ is a result of emission, sinks, physical processes and complicated chemical reactions. The explanation has no solid foundation and also needs other supporting data.

**Response:** Your suggestions are really valuable. Unfortunately, in this study we didn't conduct the chemical elements, ion, and OC/EC analysis of particulate matters and thus no available data could directly support this assumption. However, we find another reliable evidence based on the available data of the observed $PM_{2.5}$ and gaseous pollutants in our measurement to support our conclusion. To judge whether the secondary aerosol formed during the warm seasons and was further related with high $O_3$ concentrations, we chose two typical $O_3$ exceedances (OE) cases under air temperature on 10 and 12 July (OE1: 95 ppbv for average $O_3$ and 35.9 $^o$C for average T) and 10-11 August (OE2: 92.7 ppbv for average $O_3$ and 38.7 $^o$C for average T), respectively, together comparison with their nearby non-$O_3$ exceedances periods (NOE) from 7-8 July (NOE1) and 13-14 August (NOE2). Note that these data were both selected as the time period of 9:00-17:00 BLT, to reflect the photochemistry as possible. As can be seen from Table 1 below, the average $PM_{2.5}$ concentrations in OE1 and OE2 were both higher (ca. 2-4 folder) than those in NOE1 and NOE2, respectively. It suggested a significant formation of $PM_{2.5}$ in the OE event. Furthermore, to further distinguish the primary and secondary contribution to $PM_{2.5}$, we compared the ratio of the averaged $PM_{2.5}$ concentrations in OE to that in NOE events $(PM_{2.5(OE)}/ PM_{2.5(NOE)})$ with the ratios for other gaseous pollutants. If the ratio of $PM_{2.5(OE)}/PM_{2.5(NOE)}$ was comparable with that for other primary pollutants, it probably indicated that a significant contribution of primary particulate matter to the observed $PM_{2.5}$ in OE event. As clearly shown in Table 1, the ratios of $PM_{2.5(OE1)}/ PM_{2.5(NOE1)}$ and $PM_{2.5(OE2)}/ PM_{2.5(NOE2)}$ were 2.08 and 4.12, respectively, both higher than those for the other primary gaseous pollutants during these two episodes (1.20-1.61 and 1.62-2.58), indicating a significant contribution of secondary particulate matter to the observed $PM_{2.5}$ in warm seasons.

Table 1 Average concentrations of $PM_{2.5}$ and gaseous pollutants and their average ratios in the $O_3$ exceedances period on 10 and 12 July (OE1) and 10-11 August (OE2), and the nearby non-$O_3$ exceedances period from 7-8 July (NOE1) and 13-14 August (NOE2), respectively.

| Species | Same time period (9:00-17:00 BLT) | | | | | |
|---|---|---|---|---|---|---|
| | OE1* | NOE1* | OE1/NOE1 | OE2* | NOE2* | OE2/NOE2 |
| $PM_{2.5}$ | 50.65 | 24.36 | 2.08 | 41.96 | 10.17 | 4.12 |
| $O_3$ | 95.43 | 53.23 | 1.79 | 92.69 | 42.71 | 2.17 |
| $SO_2$ | 12.73 | 7.89 | 1.61 | 5.18 | 2.01 | 2.58 |
| CO | 0.46 | 0.38 | 1.20 | 0.48 | 0.30 | 1.62 |
| $NO_y$ | 35.72 | 23.95 | 1.49 | 29.30 | 16.22 | 1.81 |

*$\mu g/m^3$ unit for $PM_{2.5}$, ppmv unit for CO, and ppbv unit for the other gases, respectively

In addition, we find other simultaneous/previous observations implemented in urban Hangzhou to support our supposition. Sun et al. (2013) conducted an intensive field campaign in Hangzhou during

Sep. 2010-July 2011 and found that molar ratios of sulfate to total sulfur and nitrate to total oxidized nitrogen frequency exceeded 10%, suggesting significant effects of photochemical reactions on $PM_{2.5}$ pollution in the urban Hangzhou. Thus, secondary particulate formation may be related to high conversion rate of $SO_2$ and $NO_x$ to sulfate and nitrate under a high concentration of oxidants (Khoder, 2002; Sun et al., 2013). Note that it's necessary to implement more detailed investigations related with chemical elements, ion, and OC/EC analysis of particulate matters.

In the revised manuscript, we made corresponding corrections as mentioned above.

**References:**

Khoder, M. I.: Atmospheric conversion of sulfur dioxide to particulate sulfate and nitrogen dioxide to particulate nitrate and gaseous nitric acid in an urban area, Chemosphere, 49, 675-684, 2002.

Sun, G. J., Yao, L., Jiao, L., Shi, Y., Zhang, Q. Y., Tao, M. N., Shan, G. R., and He, Y.: Characterizing $PM_{2.5}$ pollution of a subtropical metropolitan area in China, Atmos. Climate Sci., 3, 100-110, 2013.

7. The backward trajectory and PSCF analysis is not suitable for short-lived species such as $O_3$ and is especially not suitable in urban area with high local emission. So it's strange that those clean mountain area in south of Hangzhou could have more contributions? As well as that air masses coming from open seas contained higher concentrations of $NO_x$ and $O_3$?

**Response:** We thank the referee for his/her valuable comments. We divide this comment into four questions in details:

1) The backward trajectory and PSCF analysis is not suitable for short-lived species such as $O_3$

At first, we have to clarify that this WPSCF analysis has its limitation. In principle, it's just a statistical method correlating air masses origin with the pollutants concentrations measured in a selected site. We agree with referee that the PSCF analysis used for so-called short-lived species such as $O_3$ might adds uncertainty to our results, but it will not lead to the wrong results. The method is based on the theory that those map grid cells that get much "probability" of high concentration will have an increased importance in the source area maps. A significant area will get more "probability" when a trajectory passes it again but from a slightly different direction when the length of air masses trajectory was longer than life time of the pollutant. Areas behind the source areas will have smeared concentration probability and will be mixed also with clean trajectories that have gone around the source area, thus it might arise broadening "tails" behind the significant source area with high concentration. Similar phenomenon was also found in other studies by using trajectory statistical method (Riuttanen et al., 2013).

However, this method also has significant advantage. It is a useful, widely-used, and simple way to see

where the higher concentrations (relative to a set value) come from, and thus it could represent the potential/relative source contribution fields. As mentioned in the Response 3 above, this method has been employed to elucidate the potential source contributions of particulate matters. In addition, apart from the application in investigating the potential source contributions of the trace gases such as $SO_2$, CO, and $NO_x$ (Kaiser et al., 2007; Riuttanen et al., 2013; Yu et al., 2014), it has been increasingly applied to identify the origin of $O_3$ pollution (Stohl and Kromp-Kolb, 1994; Dickerson et al., 1995; Poirot and Wishinski, 1998; Kaiser et al., 2007; Riuttanen et al., 2013; Vellingiri et al., 2016; Sharma et al., 2017), and even extended to a more complicatedly secondary pollutant of atmospheric peroxyacetyl nitrate (PAN) (Siroris and Bottenheim, 1995). As you know, $O_3$ has variable precursors and complex sink mechanisms. In fact so complex that a statistical method such as PSCF has been proven to perform better compared to deterministic trajectory based method (Schlink et al., 2003). Therefore, this method has been validated to be suitable not only for particulate matters but also for trace gases such as $O_3$, $SO_2$, CO, and $NO_x$.

2) The backward trajectory and PSCF analysis is especially not suitable in urban area with high local emission.

With respect to the applicability of this method in urban area, we have to clarify again that it just provides a general indication of the potential source probability areas **in statistical sense** and thus it's free of turbulence, dry and wet deposition, and chemical reactions (Kaiser et al., 2007). The back trajectory and PSCF method have been widely used in the analysis of atmospheric $NO_2$ and $O_3$ in urban Hangzhou (Wu et al., 2016) and the other typical urban sites in Toronto and Montreal in eastern Canada (Johnson et al., 2007), Naples in southern Italy (Riccio et al., 2007), Korea (Vellingiri et al., 2016), and New Delhi (Sharma et al., 2017). Even, this method could be used to assess the effects of transboundary ozone pollution between Ontario, Canada and New York (Brankov et al., 2003). For NRCS site, a typical urban site located in Hangzhou, it is an ideal receptor to capture the mixed signature of local emission and regional transport, with the short and long cluster-mean trajectories, respectively.

3) So it's strange that those clean mountain area in south of Hangzhou could have more contributions?

To answer the question about high contribution from the south of Hangzhou, we first view the terrain and geographical distribution in Zhejiang Province. For Zhejiang Province, its terrain inclines from southwest to northeast, and geographically many cities (i.e., Shaoxing, Jinghua, Lishui, and Quzhou city) located in the south of Hangzhou. High $O_3$ is expected to be produced in these urban regions and carried to the NRCS site by the dominant southwesterly wind. Thus these areas could act as potential source regions. In addition, we think that biogenic VOCs (BVOCs) emitted from the mountain area in the south of Hangzhou might play a certain role in the formation of local $O_3$ during the whole year except

winter. This supposition was well evidenced by seasonal and spatial distributions of O₃ volume mixing ratio (VMR) simulated by MOZART-4/GEOS-5 (See the Figure 1 below). Cleary shown in this figure, high concentrations of O₃ distributed in the south of Hangzhou including the mountain area during the whole year except winter.

[Figure]

Figure 1 Seasonal and spatial distributions of O₃ volume mixing ratio (VMR) simulated by MOZART-4/GEOS-5. The sample site is marked in pentacle.

4) As well as that air masses coming from open seas contained higher concentrations of NOₓ and O₃?

As responded to the Anonymous Referee #1, we are so sorry for the incorrectly expression "long transports from Yellow Sea, East Sea, and South Sea were also important potential sources for NOₓ and O₃" in the initial manuscript. After careful examination, we found that air masses with the seemed high WPSCF values for NOₓ were not originating far from these open seas. They were just the broadening "tails" of high values contained in the areas with intensive anthropogenic NOₓ emissions from inland well-industrialized cities. This phenomenon was also found in other studies by using trajectory statistical method (Riuttanen et al., 2013).

Similar with $NO_x$, air masses containing the high WPSCF values of $O_3$ also didn't come from the open seas. Indeed, such air masses were mostly from the offshore area of East China Sea, Yellow China Sea, or South China, respectively on southeastern Zhejiang, Jiangsu, and Fujian Province. We speculated the recirculation of pollutants by sea- and land-breeze circulations around the cities along the YRD and Hangzhou Bay which has been confirmed by Li et al. (2015, 2016b), was largely responsible for the increased concentration of $O_3$ at NRCS site. Also, such an increase in $O_3$ concentrations in urbanized coastal areas have been observed and modeled in a number of studies (Oh et al., 2006; Levy et al., 2008; Martins et al., 2012). Moreover, to further judge whether air masses came from open seas contained higher concentrations of $NO_x$ and $O_3$, we used the results of MOZART-4/GEOS-5 simulation to draw the distribution maps of $NO_x$ and $O_3$ concentrations within the identical domain (15-55 $^o$N and 105-135 $^o$E) with WPSCF analysis. As clearly seen from the Figure 2 below, high $NO_x$ mainly distributed in terrestrial regions, especially in industrialized cities, but very low $NO_x$ were found in open seas. In comparison, significantly high $O_3$ were elucidated covering the offshore regions of either East China Sea, Yellow China Sea, or South China (Fig. 1). Then, along with the seasonal cluster analysis of back trajectories from NRCS site in Hangzhou, it's well confirmed that our speculation about the contribution of the recirculation of pollutants by sea- and land-breeze circulations in the offshore area to the observed $O_3$ at NRCS site.

[Figure]

Figure 2 Seasonal and spatial distributions of NO$_x$ volume mixing ratio (VMR) simulated by MOZART-4/GEOS-5. The sample site is marked in pentacle.

In summary, we made the corresponding corrections in the revised manuscript.

**References:**

Brankov, E., Henry, R.F., Civverlo, K.L., Hao, W., Rao, S.T., Misra, P.K., Bloxam, R., and Reid, N.: Assessing the effects of transboundary ozone pollution between Ontario, Canada and New York, USA, Environ. Pollut., 123, 403-411, 2003. Dickerson, R. R., Doddridge, B. G., and Kelley, P.: Large-scale pollution of the atmosphere over the remote Atlantic Ocean: Evidence from Bermuda, J. Geophys. Res., 100, 8945-8952, 1995.

Johnson, D., Mignacca, D., Herod, D., Jutzi, D., and Miller, H.: Characterization of identification of trends in average ambient ozone and fine particulate matter levels through trajectory cluster analysis in eastern Canada, 57(8), 907-918, 2007.

Kaiser, A., Scheifinger, H., Spangl, W. G., Weiss, A., Gilge, S., Fricke, W. G., Ries, L., Cemas, D., and

Jesenovec, B.: Transport of nitrogen oxides, carbon monoxide and ozone to the Alphine global atmospheric watch stations Jungfraujoch (Switzerland), Zugspitze and Hohenpeissenberg (Germany), Sonnblick (Austria) and Mt. Krvavec (Slovenia), Atmos. Environ., 41, 9273-9287, 2007.

Poirot, R. L. and Wishinski, P. R.: Long-term ozone trajectory climatology for the eastern US, 2, Results, 98-TP43.06 (A615), paper presented at the 91st Air and Water Management Association Annual Meeting and Exhibition, San Diego, Calif., 14-18 June, 1998.

Riccio, A., Giunta, G., and Chianese, E.: The application of a trajectory classification precodure to interpret air pollution measurements in the urban are of Naples (Southern Italy), Sci. Total. Environ., 376, 198-214, 2007.

Riuttanen, L., Hulkkonen, M., Dal Maso, M., Junninen, H., and Kulmala, M.: Trajectory analysis of atmospheric transport of fine particles, $SO_2$, $NO_x$ and $O_3$ to the SMEAR II station in Finland in 1996-2008, Atmos. Chem. Phys., 13, 2153-2164, 2013.

Schlink, U., Dorling, S., Pelikan, E., Nunnari, G., Cawley, G., Junninen, H., Greig, A., Foxall, R., Eben, K., Chatterton, T., Vondracek, J., Richter, M., Dostal, M., Bertucco, L., Kolehmainen, M., and Doyle, M.: A rigorous inter-comparison of ground-level ozone predictions, Atmos. Environ., 37(23), 3237-3253, 2003.

Sharma, A., Mandal, T. K., Sharma, S. K., Shukla, D. K., and Singh, S.: Relationship of surface ozone with its precursors, particulate matter and meteorology over Delhi, J. Atmos. Chem., 74(4), 451-474, 2017.

Siroris, A. and Bottenheim, J. W.: Use of backward trajectories to interpret the 5-year record of PAN and $O_3$ ambient air concentrations at Kejimkujik National Park, Nova Scotia, J. Geophys. Res., 100, 2867-2881, 1995.

Stohl, A. and Kromp-Kolb, H.: Origin of ozone in Vienna and surroundings, Austria, Atmos. Environ., 28, 1255-1266, 1994

Vellingiri K., Kim, K. H., Lim, J. M., Lee, J. H., Ma, C. J., Jeon, B. H., Sohn, J. R., Kumar, P., and Kang, C. H.: Identification of nitrogen dioxide and ozone source regions for an urban area in Korea using back trajectory analysis, Atmos. Res., 176-177, 212-221, 2016.

Wu, J., Xu, C., Wang, Q. Z., and Cheng, W.: Potential sources and formations of the $PM_{2.5}$ pollution in urban Hangzhou, Atmosphere, 7(100), 1-15, 2016.

Yu, S. C., Zhang, Q. Y., Yan, R. C., Wang, S., Li, P. F., Chen, B. X., Liu, W. P., and Zhang, X. Y.: Origin of air pollution during a weekly heavy haze episode in Hangzhou, China, Environ. Chem. Lett., 12, 543-550, 2014.

---

## Referee Report (RR1)

The authors have addressed all my major concerns about the original submission. Now the manuscript can be accepted after the following technical corrections are made.

1. In the response to the 9[th] comment: when describing the potential overestimation of $NO_2$ by the MoO conversion method, the authors may consider to refer to Xu et al. (2013), who have evaluated the performance of this traditional catalytic conversion method in urban, rural and remote environments.

   Xu, Z., Wang, T., Xue, L.K., Louie, P.K.K., Luk, C.W.Y., Gao, J., Wang, S.L., Chai, F.H., Wang, W.X., 2013. Evaluating the uncertainties of thermal catalytic conversion in measuring atmospheric nitrogen dioxide at four differently polluted sites in China. Atmospheric Environment 76, 221-226.

2. In the response to the 16[th] comment, "In summer (Fig. 2f), an abnormally high level of Ox was found in winter with low $O_3$": this sentence is hard to follow. Please revise it.